# STAG2 regulates interferon signaling in melanoma via enhancer loop reprogramming

Zhaowei Chu[1,15,16], Lei Gu[2,16✉], Yeguang Hu[1,16], Xiaoyang Zhang [3,16], Man Li[1], Jiajia Chen [4,5], Da Teng[1], Man Huang [6], Che-Hung Shen[7], Li Cai[8], Toshimi Yoshida [1], Yifeng Qi [9], Zhixin Niu[2], Austin Feng[1], Songmei Geng[10], Dennie T. Frederick [11], Emma Specht[11], Adriano Piris[12], Ryan J. Sullivan [11], Keith T. Flaherty [11], Genevieve M. Boland[13], Katia Georgopoulos [1], David Liu[4,5], Yang Shi [6,14] & Bin Zheng [1✉]

The cohesin complex participates in the organization of 3D genome through generating and maintaining DNA loops. Stromal antigen 2 (STAG2), a core subunit of the cohesin complex, is frequently mutated in various cancers. However, the impact of STAG2 inactivation on 3D genome organization, especially the long-range enhancer-promoter contacts and subsequent gene expression control in cancer, remains poorly understood. Here we show that depletion of STAG2 in melanoma cells leads to expansion of topologically associating domains (TADs) and enhances the formation of acetylated histone H3 lysine 27 (H3K27ac)-associated DNA loops at sites where binding of STAG2 is switched to its paralog STAG1. We further identify Interferon Regulatory Factor 9 (IRF9) as a major direct target of STAG2 in melanoma cells via integrated RNA-seq, STAG2 ChIP-seq and H3K27ac HiChIP analyses. We demonstrate that loss of STAG2 activates IRF9 through modulating the 3D genome organization, which in turn enhances type I interferon signaling and increases the expression of PD-L1. Our findings not only establish a previously unknown role of the STAG2 to STAG1 switch in 3D genome organization, but also reveal a functional link between STAG2 and interferon signaling in cancer cells, which may enhance the immune evasion potential in STAG2-mutant cancer.

[1] Cutaneous Biology Research Center, Massachusetts General Hospital and Harvard Medical School, Charlestown, Massachusetts, USA. [2] Epigenetics Laboratory, Max Planck Institute for Heart and Lung Research, 61231 Bad Nauheim, Germany. [3] State Key Laboratory of Genetic Engineering, School of Life Sciences, Fudan University, Shanghai, China. [4] Department of Medical Oncology, Dana-Farber Cancer Institute, Boston, MA, USA. [5] Broad Institute of MIT and Harvard, Cambridge, MA, USA. [6] Division of Newborn Medicine, Boston Children's Hospital, Harvard Medical School, Boston, MA, USA. [7] National Institute of Cancer Research, National Health Research Institutes, Tainan, Taiwan. [8] Department of Cancer Biology, University of Texas MD Anderson Cancer Center, Houston, TX, USA. [9] Department of Chemistry, Massachusetts Institute of Technology, Cambridge, MA 02139, USA. [10] Department of Dermatology, The Second Hospital Affiliated to Xi'an Jiaotong University, Xi'an, Shaanxi, China. [11] Department of Medicine, Massachusetts General Hospital Cancer Center, Boston, MA, USA. [12] Department of Dermatology, Brigham and Women's Hospital and Harvard Medical School, Boston, MA, USA. [13] Department of Surgical Oncology, Massachusetts General Hospital, Boston, MA, USA. [14] Ludwig Institute for Cancer Research, Oxford University, Oxford, United Kingdom. [15] Present address: Department of Dermatology, Northwest Hospital, The Second Hospital Affiliated to Xi'an Jiaotong University, Xi'an, Shaanxi, China. [16] These authors contributed equally: Zhaowei Chu, Lei Gu, Yeguang Hu, Xiaoyang Zhang. ✉email: Lei.Gu@mpi-bn.mpg.de; bin.zheng@cbrc2.mgh.harvard.edu

STAG2, or its paralog STAG1, interacts with a tripartite ring-like structure composed of SMC1A, SMC3, and RAD21, to form the core of the cohesin complex[1,2]. STAG2 and STAG1 share high sequence similarities and can partially compensate for each other's function[2,3]. However, mutations in STAG2, but not STAG1, occur frequently in various cancers, suggesting a unique role of STAG2 in cancer biology[2,4,5]. We recently reported that inactivation of STAG2, but not STAG1, results in resistance to BRAF pathway inhibition in melanoma[6]. Knockdown of STAG2 in melanoma cells did not apparently affect cell cycle progression, apoptosis, or growth of xenograft tumors in nude mice, but significantly decreased the sensitivities of melanoma cells to inhibition of BRAF and/or MEK in both culture cells and xenograft mouse models. Mechanistically, we showed that loss of STAG2 inhibits CTCF-mediated expression of dual-specificity phosphatase 6 (DUSP6), a negative regulator of the BRAF-MEK-ERK signaling pathway, leading to elevated ERK activity[6]. However, the potential tumor suppressor function of STAG2 in melanoma and the underlying mechanism remains unclear.

Here, we show that IRF9, a key component in the type I interferon signaling pathway, is a direct target of STAG2 in melanoma through integrated analyses of RNA-Seq, ChIP-seq, and H3K27ac HiChIP. Loss of STAG2 in melanoma results in IRF9 activation, which in turn upregulates PD-L1 expression in cancer cells, suggesting a potential tumor suppressor function of STAG2 in immune evasion.

## Results

**Loss of STAG2 disrupts TAD organization in melanoma**. The cohesin complex regulates the organization of the 3D genome through generating and maintaining DNA loops[7–10]. To characterize the role of STAG2 in 3D genome organization in melanoma cells, we carried out ChIP-seq against STAG2, STAG1, SMC1A, and CTCF as well as Hi-C analyses (Supplementary Fig. 1a–c and Supplementary Data 1) in M14 melanoma cells stably expressing inducible shRNA of STAG2[6]. In total, 79,115 peaks were identified in STAG2 WT samples. Upon STAG2 KD, we identified 32,647 peaks with significant loss of binding activity and 998 with a gain of binding activity. As for the STAG1 binding profile, there were 285 peaks showing significant loss of binding activity and 30,621 peaks with significant gain of binding activity upon STAG2 KD. As expected, the genomic distributions of differential peaks between STAG1 and STAG2 were very similar. The majority of the differential peaks were located in Introns and Intergenic regions (Supplementary Fig. 1d). Consistent with previous reports[11,12], we found that loss of STAG2 binding caused by knockdown (KD) of STAG2 was frequently accompanied by increased STAG1 binding (~41% of sites) (Fig. 1a–c and Supplementary Fig. 2a–d), suggesting a switch from STAG2 to STAG1 at these sites. Although we did not observe widespread changes of CTCF binding upon STAG2 knockdown (Fig. 1a), the STAG2 to STAG1 switch events preferentially occur at STAG2 sites that are also bound by SMC1A and CTCF (Fig. 1c). Further analysis of Hi-C data revealed slightly reduced compartmentalization (Supplementary Fig. 3a, b), elevated contact probability (Fig. 1d and Supplementary Fig. 3c–f), and significantly increased TAD size on average (Fig. 1e, f and Supplementary Fig. 4a–c, $p = 3.488e-13$, Mann–Whitney $U$-test) upon STAG2 KD. In total, we identified 3789 and 3496 TADs from WT and KD with 100 Kb bin size, respectively. We then expanded a given TAD's left and right genomic coordinates with 10 kb on both directions and defined them as its two boundary regions for downstream analysis. If two boundary regions were overlapped, they would be merged as a big boundary. Through this approach,

6260 and 5761 boundaries were identified from WT and KD, respectively. Next, we determined the overlap of the boundaries between WT and KD and separated them as 4380 stable boundaries (SB) and 3281 variable boundaries (VB). We observed that 18% of changed TADs with variable boundaries had completely new boundaries being formed, 56% of them had neighboring TADs merged and 26% of them had either boundary shifted (Supplementary Fig. 5a). In order to further understand whether those variable boundaries of TADs were truly lost/gained or there were just only changes in boundary strength, we next calculated insulation scores to quantitatively measure the strength of TAD boundary[13]. We found that the insulation levels at stable boundaries were significantly higher than those at variable boundaries (Supplementary Fig. 5b, c). We further analyzed cohesin and CTCF binding at the TAD boundaries and found that the strength of cohesin and CTCF binding were weaker at boundaries of variable TADs that changed upon STAG2 KD than the boundaries of stable TADs (Fig. 1g), suggesting that TADs with weaker cohesin or CTCF binding sites at the boundaries are more susceptible to STAG2 knockdown. We further carried out a meta-analysis of gene expression with a 10 kb sliding window across all variable and stable TADs, including their boundaries, and observed remarkable changes of gene expression at the boundaries of variable TADs, but not the stable TADs (Supplementary Fig. 4g). To further investigate whether there were any different genomic patterns between the switch and non-switch sites, we calculated their associations with genomic positions of TAD domains, boundaries, enhancers, and promoters. We found that switch sites were enriched in TAD boundaries but depleted in TAD domains and enhancers, and no significant differences were observed in promoters (Supplementary Table 1). Interestingly, among subgroups of variable TADs (Supplementary Fig. 4d–f), there is a remarkable switch of STAG2 to STAG1 occurred at boundaries of expanded TADs, but not at those of shrinked TADs (Fig. 1h). These findings together suggest that, upon STAG2 inactivation in melanoma cells, the switch of STAG2 to STAG1 binding at TAD boundaries could be associated with the expansion of TADs, which may contribute to TAD enlargement and subsequently increased frequency of spatial interaction.

**STAG2 modulates the landscape of H3K27ac-associated loops**. Next, we investigated whether the alterations in TAD boundaries caused by STAG2 loss are accompanied by changes in the promoter–enhancer interactions. We performed HiChIP analyses[14] of H3K27ac, an active enhancer- and promoter-associated histone marker in the same M14 cell line, and compared the H3K27ac-associated interaction landscape with and without STAG2 knockdown (Fig. 2a, Supplementary Fig. 6a, and Supplementary Data 1). In total, we identified 9115 enhanced loops and 11327 impaired loops upon STAG2 knockdown. The average length of the enhanced loops is significantly longer than the impaired ones (Supplementary Fig. 6b, $p < 2.2e-16$, Mann–Whitney $U$-test), which is consistent with the Hi-C results that STAG2 knockdown usually causes enhanced contact probabilities for long-range interactions, rather than short-range interactions (Supplementary Fig. 3a). In addition, we found that the majority of the altered loops (~80%) occurred within TADs, rather than across the TAD boundaries, and we did not observe significant differences between enhanced and impaired loops, either within or across TADs (Fig. 2b). Further analysis at anchors of the enhanced and impaired loops revealed a distinct spectrum of transcription factor motifs (Fig. 2c). We next categorized H3K27ac-associated loops into three groups: promoter–promoter interaction (PP), promoter–enhancer interaction (PE), and enhancer–enhancer interaction (EE), and found no

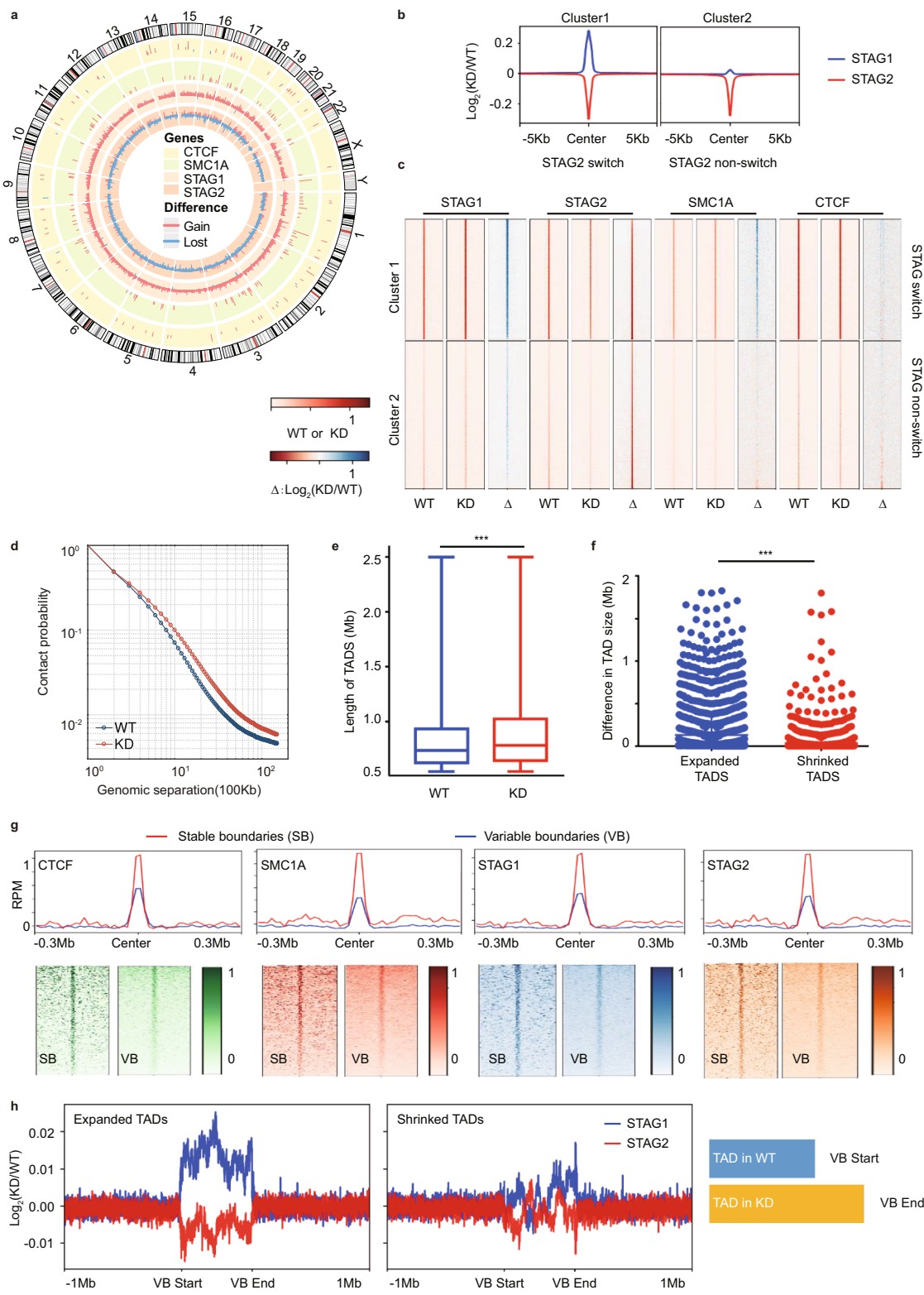

apparent shifts in the percentages of these three groups between WT and KD cells, nor between enhanced and impaired loops (Supplementary Fig. 6c, d). Consistent with recent findings in other cell lineages[12,15,16], we also found STAG2 KD in M14 cells led to significant gains of super-enhancers (Fig. 2d and Supplementary Fig. 6e, f). Metaplots of H3K27ac across super-enhancers further revealed that the changes of super-enhancers upon STAG2 knockdown reflected corresponding changes in H3K27ac levels

(Supplementary Fig. 6g). Furthermore, we analyzed patterns of H3K27ac binding and H3K27ac-associated DNA loop at the sites of STAG2 to STAG1 switch versus non-switch sites. We extracted average H3K27ac signals for each switched and non-switched site and compared the mean of the H3K27ac at these sites between WT and KD. Interestingly, we observed significant gains of H3K27ac signal upon STAG2 KD at switch sites ($p < 2.2e-16$, Wilcox test), but not at non-switch sites ($p = 0.1368$, Wilcox test) (Fig. 2e),

**Fig. 1 Loss of STAG2 affects 3D genome structure. a** Circos plot for differential peaks of CTCF, SMC1A, STAG1, and STAG2 binding upon STAG2 knockdown in M14 cells. Red bar represents gain of binding and blue bar represents loss of binding. **b** Distinct profiles of changes in STAG1 binding at the loss of STAG2 binding sites upon STAG2 knockdown in M14 cells. The STAG switch and non-switch groups are identified by k-means clustering and show significant gain or nearly no gain in STAG1 binding at the loss of STAG2 binding sites, respectively. **c** Heatmap of normalized ChIP-seq signals for STAG1, STAG2, SMC1A, and CTCF in M14 cells with (KD) and without (WT) STAG2 shRNA knockdown and their changes (Δ), as defined by $\log_2(KD/WT)$, in both STAG switch (cluster 1) and non-switch groups (cluster 2). **d** Average contact probability at a different genomic distance for KD and WT, using 100 kb as window size. **e** Average TAD length is significantly different between STAG2 WT ($n = 3789$ TADs) and KD ($n = 3496$ TADs). The box plot is defined by bounds at the 25th percentile and 75th percentile, center at 50th percentile, the minima and maxima are at the 10th percentile and 90th percentile. *P* value is 3.488e-13 and is based on a two-sided Wilcox test. **f** Difference in TAD size for expanded TADs ($n = 1125$ TADs) is significantly longer than those of shrinked TADs ($n = 1007$ TADs). *P* value is 2.312e-17 and is based on a two-sided Wilcox test. **g** CTCF, SMC1A, STAG1, and STAG2 profiles at stable and variable TAD boundaries in M14 STAG2 WT cells. RPM reads per million. **h** Remarkable STAG2 to STAG1 switch occurs at boundaries of expanded TADs but not shrinked TAD upon STAG2 knockdown. The schematic diagram depicts that the variable boundary (VB) start site is defined as the site in WT, and the related end site as the site in KD. Statistical significance is determined as: ns *P* > 0.05; *P* < 0.05; **P* < 0.01; ***P* < 0.001.

indicative of increased regulatory activity. We also found a remarkable increase of H3K27ac signal at anchors of enhanced loops (Fig. 2f). More importantly, while anchors of both the enhanced and impaired H3K27ac loops showed switches from STAG2 to STAG1, the degree of the switch is markedly different (Fig. 2g). Anchors of the impaired loops showed a significantly lower level of compensation, compared to those of the enhanced loops (Fig. 2g). Together, these data suggest that the switch of STAG2 to STAG1 is associated with the increase of H3K27ac binding and enhanced H3K27ac loop.

**IRF9 is a key direct target of STAG2 in melanoma**. We next carried out an integrated analysis of RNA-seq, STAG2 ChIP-Seq, and H3K27ac HiChIP data to identify direct targets of STAG2 based on if the genes (1) are differentially expressed, (2) have reduced STAG2 binding at promoters, (3) are associated with a differential H3K27ac loop upon STAG2 knockdown in M14 cells (Fig. 3a). We found 148 genes as potential direct targets of STAG2 in these cells. Gene set enrichment analysis revealed interferonα (IFNα) response and IFNγ response as the top two most significantly enriched Hallmark pathways induced by STAG2 knockdown (Fig. 3b and Supplementary Fig. 7a–c). We focused on characterizing IRF9 as a direct target of STAG2 in melanoma cells (Fig. 3c), because that IRF9 is a central transcription factor in type I interferon signaling pathway[17,18] and that motifs of several IRF family members were found to be enriched in the enhanced H3K27ac loop anchors (Fig. 2d). Hi-C and H3K27ac HiChIP analyses indicated that STAG2 loss leads to an expansion of a downstream TAD of IRF9 to encompass IRF9 and the formation of a new H3K27ac-associated interaction between IRF9 promoter and a distal enhancer within the enlarged TAD (Fig. 3d and Supplementary Fig. 8a–d). We confirmed knockdown of STAG2 (Supplementary Fig. 7l–n) increased levels of IRF9 expression in multiple melanoma cell lines (Fig. 3e). We also observed increased expression of several key targets of type I IFN pathway, such as ISG15, USP18, and IRF7, upon STAG2 knockdown (Fig. 3f), confirming increased mRNA expression of these genes observed in RNA-seq analysis. In contrast, shRNA knockdown of STAG1 or CTCF apparently did not affect the protein levels of IRF9 in M14 cells (Supplementary Fig. 7d, e). Expression of an shRNA-refractory mutant of STAG2 rescued the effects of STAG2 KD on IRF9 expression in M14 cells (Supplementary Fig. 7f). Conversely, ectopic expression of STAG2 in WM902BR cells that bear loss-of-function mutation of STAG2[6] reduced IRF9 expression (Supplementary Fig. 7i–k), further supporting a repressive role of STAG2 in IRF9 expression. Importantly, CRISPR knockout of IRF9 (Supplementary Fig. 7o) in M14 cells abolished the induction of these Type I IFN response genes by STAG2 knockdown or IFNβ treatment (Fig. 3f),

supporting that STAG2 represses the type I IFN signaling pathway via suppressing IRF9 expression.

**STAG2 regulates PD-L1 expression in melanoma via IRF9**. Since IFN signaling and IRF9 in particular play critical roles in regulating the expression of PD-L1 in cancer cells[19,20], we examined whether STAG2 modulates the expression of PD-L1 in melanoma cells. We found that knockdown of STAG2 increased both mRNA and protein levels, as well as the surface expression of PD-L1 in melanoma cells (Fig. 4a–d and Supplementary Fig. 7f–h). As expected, CRISPR knockout of IRF9 reversed the induction of PD-L1 protein level in M14 cells (Fig. 4e).

We next investigated whether the regulation of 3D genome organization at the IRF9 locus by STAG2 mediates its effects on IRF9 and PD-L1 expression. We carried out CRISPR-mediated deletion of the STAG2 binding site at the boundary of the TAD downstream of IRF9 that was expanded upon STAG2 KD (Fig. 3d and Supplementary Fig. 9a, b) in M14 cells. The mRNA levels of both IRF9 and PD-L1, as well as the surface expression of PD-L1 increased in three independent clones of CRISPR deletion (Fig. 4f–h). As a negative control, deletion of another STAG2 binding site in the area, but not at the boundary (Supplementary Fig. 9a), did not affect the expression of IRF9 mRNA (Supplementary Fig. 9c). Moreover, we applied the KRAB-dCas9 system[21] to repress the putative distal enhancer of IRF9 (Fig. 3d) identified in our H3K27ac HiChIP analysis. STAG2 knockdown fails to induce IRF9 expression, as well as subsequent PD-L1 expression, after the repression of the enhancer (Fig. 4i). These results together support that STAG2 controls the expression of IRF9 via expanding the TAD boundary and enhancing an H3K27ac-associated enhancer–promoter loop (Fig. 4o).

Finally, to validate the association between STAG2, IRF9, and PD-L1 in clinical tumor samples, we analyzed skin cutaneous melanoma (SKCM) bulk RNA-seq data ($n = 473$ tumors) from the cancer genome atlas (TCGA)[22]. As expected, we found that STAG2 was negatively correlated with IRF9 (Spearman, $r = -0.21$, $p = 2.4e-06$) (Fig. 4j) and IRF9 was positively correlated with PD-L1 (Spearman, $r = 0.5$, $p < 2.26-16$) (Fig. 4k). Surprisingly, STAG2 was positively correlated with PD-L1 expression in the unadjusted analysis (Supplementary Fig. 10a), but in the multivariate analysis was an independent negative predictor of PD-L1 expression after adjusting for STAT1 (i.e., a non-IRF9 regulator of PD-L1 expression) (Fig. 4l, Supplementary Fig. 10a, and Supplementary Table 2). We then examined the association of *STAG2* expression with response to PD-1 immune checkpoint blockade in a large cohort of metastatic melanoma patients. Lower STAG2 mRNA expression (median split) was associated with increased response to anti-PD-1 therapy (two-sided Fisher's exact test, $p = 0.048$; OR = 2.29 95% CI 0.97–5.52)

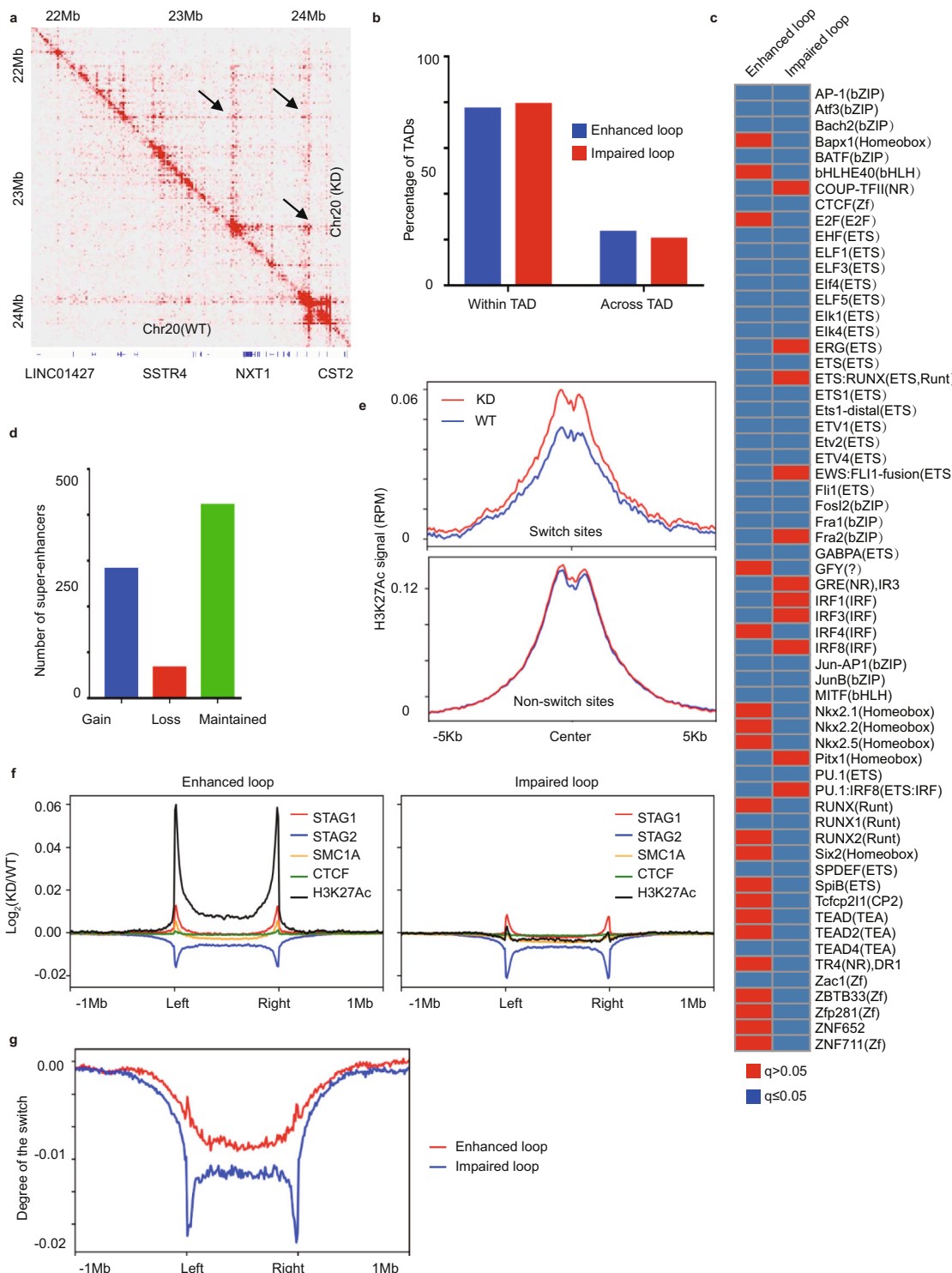

**Fig. 2 STAG2 modulates the landscape of H3K27ac-associated loops. a** Heatmap representation of differential H3K27Ac-associated loops between M14 cells with (KD) and without (WT) STAG2 shRNA knockdown as indicated by black arrows. **b** Majority of loop changes upon STAG2 knockdown occurs within TADs. **c** Motifs of transcription factors enriched at anchors of enhanced and impaired loops. **d** Numbers of gained, lost, and maintained super-enhancers upon STAG2 knockdown in M14 cells. **e** Gain of H3K27ac signal at the STAG2 to STAG1 switch sites, but not non-switch sites, upon STAG2 knockdown. **f** Differential CTCF, SMC1A, STAG1, STAG2, and H3K27ac profiles at anchors of enhanced and impaired loops upon STAG2 knockdown. **g** The relative gain of STAG1 in compensation to the loss of STAG2 is significantly higher at anchors of enhanced loops than those of impaired loops. The degree of the switch is calculated as the sum of log$_2$(KD/WT) for STAG1 and STAG2 (method).

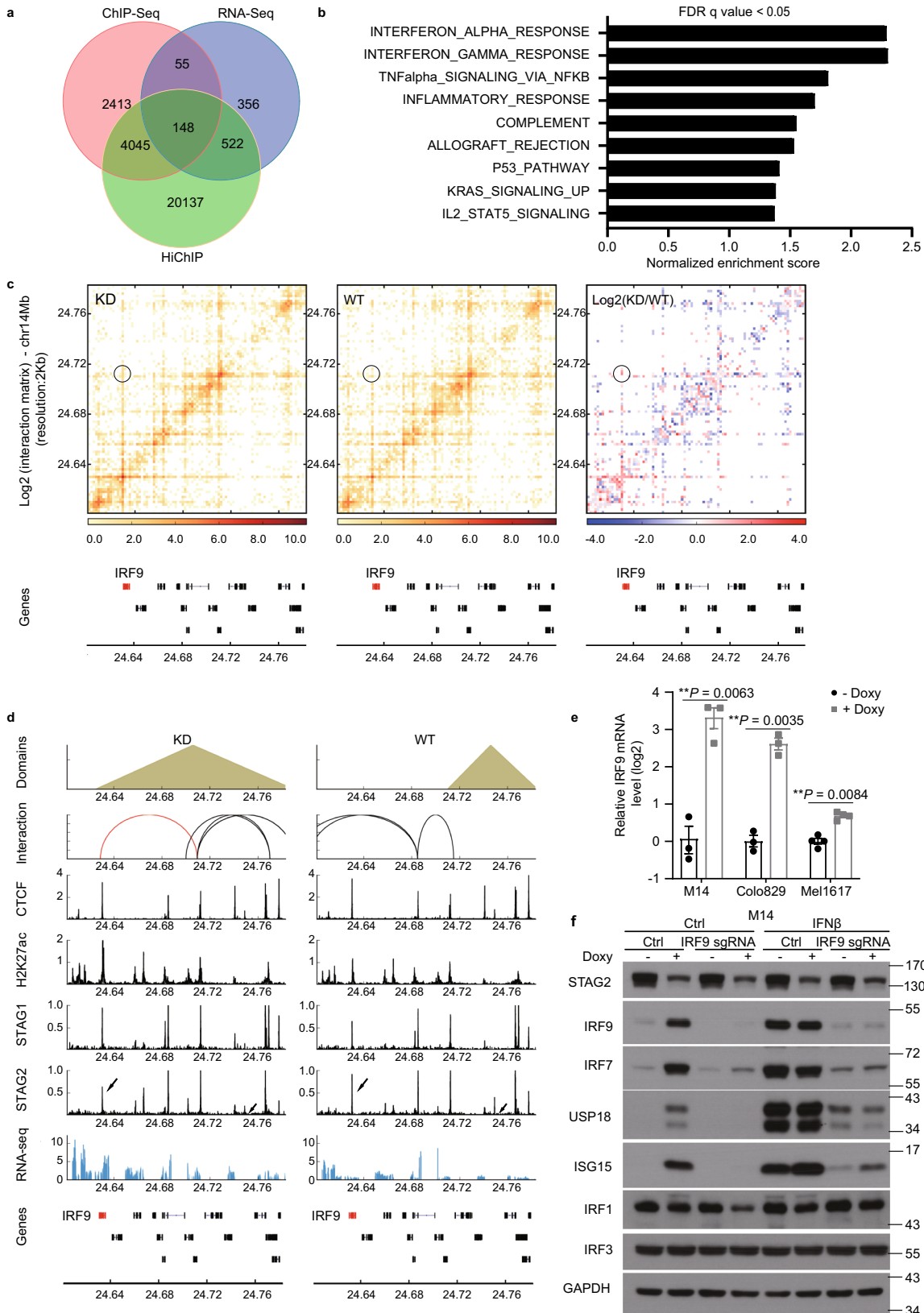

(Fig. 4m and Supplementary Fig. 10b) and longer progression-free survival (PFS) (two-sided KM log-rank test, $p = 0.032$) (Fig. 4n), consistent with the hypothesis that tumors with upregulation of PD-L1 by STAG2 inactivation may be susceptible to blockade of the PD-L1/PD-1 immune checkpoint pathway.

## Discussion

Our findings here provide further insight into the role of STAG2 in the 3D genome organization and transcriptional control. The regulation of promoter–enhancer interactions and TAD formation by the cohesin complex is well established[23–31].

**Fig. 3 IRF9 is a key direct target of STAG2 in melanoma. a** Venn diagram illustrates targets of STAG2 identified through integrated H3K27ac HiChIP, RNA-seq, and STAG2 ChIP-seq analyses in M14 cells. **b** GSEA of RNA-seq data ($n = 3$) reveals enriched pathways induced by STAG2 KD in M14 cells. **c** STAG2 knockdown led to the formation of a new H3K27ac-associated interaction between IRF9 promoter and a distal enhancer, as revealed in H3K27ac HiChIP analysis ($n = 2$). **d** Hi-C, H3K27ac HiChIP, ChIP-seq, and RNA-seq profiles at the IRF9 locus in M14 cells with (KD) and without (WT) STAG2 knockdown. An H3K27ac-associated loop anchored at the IRF9 promoter is highlighted in red. **e** qPCR analysis of IRF9 mRNA levels in human melanoma cell line M14 ($n = 3$), Colo829 ($n = 3$), and Mel1617 ($n = 4$) with doxy-inducible STAG2 shRNA. Data were presented as mean ± SEM. Two-tailed ratio paired $t$-tests, **$P < 0.01$. **f** M14 cells with doxy-inducible STAG2 shRNA were transfected with ctrl or IRF9 sgRNA. Cells were cultured in the presence or absence of doxycycline for 5 days and stimulated with or without IFNβ (500U/ml) for 24 h before lysates were used for immunoblotting with indicated antibodies. ($n = 3$). For all panels, $n$ values indicate the number of biologically independent samples.

However, the effects of STAG2 depletion on the landscape of enhancer–promoter interactions in mammalian cells remain to be characterized. Our H3K27ac HiChIP analysis revealed that the switch of STAG2 to STAG1 upon STAG2 inactivation often leads to a gain in the H3K27ac at the switch site and promotes the formation of new enhancer–promoter loops (Fig. 2g), as clearly manifested in the case of IRF9 (Figs. 3d and 4o). These findings argue against a direct role of STAG2 on regulating gene expression, as previously proposed[12,32,33], and offer an important clue to the mechanism by which STAG2 modulates gene expression via modulating H3K27ac-associated DNA loops, which may contribute to dysregulated gene expression in STAG2 mutant cancer.

Moreover, inactivation of STAG2, unlike other cohesin complex proteins, was previously reported to only have no to minimal effects on TADs in MCF10A human transformed breast epithelial cells[12], mouse embryonic stem cells[16], mouse HSPCs[11], HCT116 human colorectal carcinoma cells[32], or RT112 bladder cancer cells[39], which led to the hypothesis that STAG2 may not be required for the maintenance of TADs, probably due to compensation of STAG1[2]. However, our in-depth analysis on the TAD boundaries reveal that the switch of STAG2 to STAG1 at the TAD boundaries could be actually associated with the expansion of TADs upon STAG2 knockdown in melanoma cells (Fig. 1g). Due to the limitation in the scope of our Hi-C analysis, we acknowledge the possibility that a mix of technical and biological variations could contribute to the changes in the TAD organization that we observed. Although the mechanism underlying the effect of STAG2 to STAG1 switch on TAD expansion is currently unknown, it is noteworthy that STAG2 was shown to have a higher affinity than STAG1 for WAPL[16], a cohesin dissociation factor and key regulator of loop extrusion and TAD formation[34].

Lastly, our identification of IRF9 as a key target of STAG2 in the context of 3D genome organization in melanoma cells raises the possibility that upregulation of type I interferon signaling may contribute to the tumor suppressor function of STAG2. Although the effects of type I interferon signaling in tumor biology are context-dependent and could be either pro- or anti-tumor[18], activation of tumor-intrinsic type I interferon signaling has recently been demonstrated to induce tumor immune evasion and promote tumor formation and progression, through induction of immune checkpoint protein PD-L1 expression and recruitment of myeloid-derived suppressor cells in the tumor microenvironment[35–38]. Further investigation on the regulation of tumor immune evasion by STAG2 will provide insight into the tumor suppression mechanism of the cohesin complex, one of the most frequently mutated protein complexes in cancer[2].

## Methods

**Cell culture**. All cell lines used in this study and their culture conditions have been described previously except for Colo829[6]. To induce the knockdown of STAG2, M14 cell line expressing STAG2 doxycycline-inducible shRNA#60 were treated with (KD) or without (WT) doxycycline (5 ug/ml) for 5 days. Colo829 cells were purchased from ATCC and grown in DMEM supplemented with 10% FBS and 1%

penicillin- streptomycin-glutamine. Recombinant interferon-β protein (IF014) was purchased from Millipore. For lentiviral infection, HEK293T cells were co-transfected with lentiviral vectors, psPAX2 and pCMV-VSV-G to produce the virus. Target cells were infected with lentiviral particles with polybrene as previously described[6]. Stable populations were selected and maintained by selection with 100 ug/ml hygromycin, 4 ug/ml blasticidin, or 1 ug/ml puromycin.

**ChIP-seq**. ChIP-seq experiments were performed as previously described[40]. Briefly, cells were fixed with 1% formaldehyde at RT for 10 min, lysed and sonicated in RIPA buffer containing 0.2% SDS (50 mM HEPES pH 7.9, 140 mM NaCl, 1 mM EDTA, 1% Triton X-100, 0.1% Na-deoxycholate, 0.2% SDS, 0.5 mM PMSF, and 1x Protease inhibitor cocktail (Roche). Chromatin was cleared by centrifugation at 20,000×g for 10 min and incubated with 2–10 μg of antibodies pre-bound to Dynabeads Protein G (Life technologies). The antibodies used here were specific for STAG1 (Abcam, ab4457, lot: GR279696-4), STAG2 (Abcam, ab4464, Lot: GR271549-1), SMC1A (Bethyl, A300-055A, lot 5), and CTCF (Cell Signaling Tech, 2899 s, lot 2). Purified ChIP DNA was end-repaired, end adenylated, and ligated with Illumina Truseq indexed adapters. The ligated DNA was purified with AMPure XP beads (Beckman Coulter) and amplified with KAPA HiFi DNA Polymerase (KAPA Biosystems) for 8 to 13 cycles. After amplification, the library DNA was size-selected with AMPurex XP beads to 200–600 bp range, and the purified libraries were multiplexed for sequencing at the Center for Computational and Integrative Biology (CCIB) DNA Core at MGH.

**ChIP-seq data processing**. Raw reads were mapped to hg19 using bowtie2 (version 2.2.9). Low-quality reads and PCR duplicates were filtered by samtools (version: 1.9) and picard toolkit (version: 2.8.0). Peaks were called by MACS2 (version: 2.1.1.20160309) against input controls. Super-enhancer was identified as previously described[41]. Briefly, H3K27ac peaks within 12.5 kb of one another were stitched together and ranked by the H3K27ac enrichment. The stitched enhancers are plotted based on their ranking and the ones past the point where the slope is equal to 1 are called as super-enhancers. Data were visualized by the R package circlize[42]. To evaluate whether the overlap sites between loss of STAG2 and gain of STAG1 (switch mode) is random or nonrandom, we compared observed overlap sites to the simulated overlap sites. The empirical $P$ value is determined by 1000 permutations. Motif analysis on the switch and non-switch sites was performed by HOMER as standard set up[43].

**In situ Hi-C**. M14 cells (~$4 \times 10^6$) were crosslinked with 1% formaldehyde at RT for 10 min. Fixation was quenched by the addition of 0.125 M glycine for 10 min. In situ Hi-C was performed following the protocol by Rao et al.[44]. Briefly, nuclei were permeabilized, and DNA was digested overnight with 100 U MboI (New England BioLabs). The ends of the restriction fragments were labeled using biotin-14-dATP (Life Technologies) and then ligated in 1.2 mL final volume. After reversal of crosslinks ligated DNA was purified and sheared to a length of ~400 bp with a Covaris E220 evolution instrument (Covaris, Woburn, MA). Ligation junctions were then pulled down with streptavidin beads, and DNA fragments were end-repaired, dA-tailed, and Illumina adapters ligated. Libraries were produced by eight cycles of PCR amplification with KAPA Hifi DNA polymerase (Roche). The Hi-C libraries were size-selected with AMPure XP beads and sequenced at Quick Biology Inc. (Pasadena, CA).

Hi-C paired-end reads were aligned to hg19 using Hi-C Pro pipeline[45] and visualized by the HiCPlotter[46]. After mapping, uninformative reads were removed by HOMER "makeTagDirectory update" command[43]. After QC, HOMER "analyzeHiC" command was used to generate and normalize contact matrices with 1 Mb resolution. Chromatin compartment analysis (PCA) was applied using HOMER runHiCpca.pl and "analyzeHiC" command was used to generate DLR (Distal-to-Local [log2] Ratio) and ICF (Interchromosomal Fraction of Interactions) scores from Hi-C data.

To calculate the average contact probabilities at different genomic distances for M14 STAG2 WT and KD cells, we first calculated the average Hi-C contacts at different genomic distances and then normalized by the mean value of the contacts in the first off-diagonal of Hi-C map under different resolutions (100 kb, 250 kb, 500 kb, or 1 Mb as indicated)[47]. We carried out this process for each chromosome individually and then averaged over all chromosomes to generate the averaged

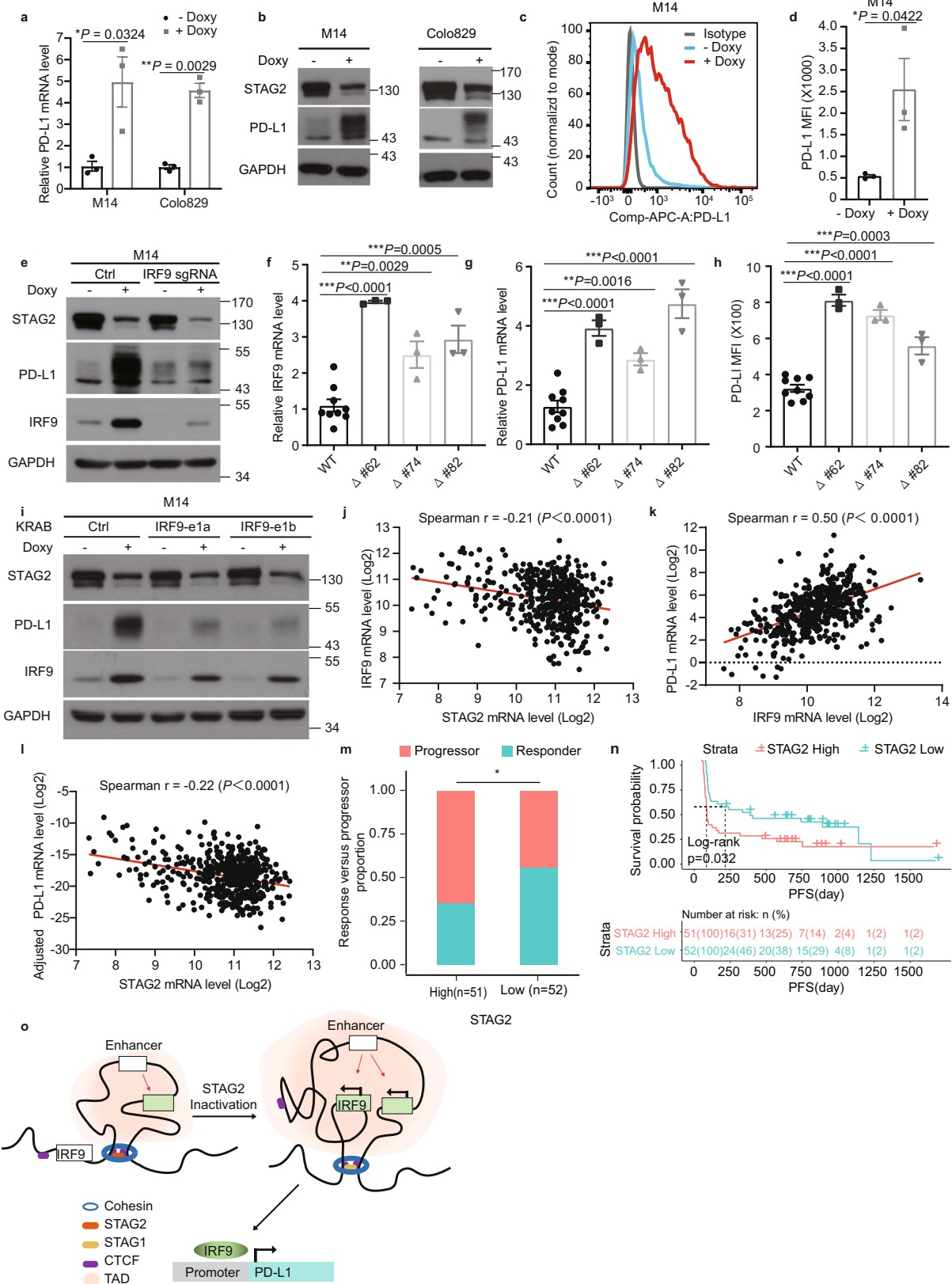

data. With this probability normalization, the genomic segments that are within the resolution of the Hi-C map along the sequence will on average have a contact probability of 1, which enables a direct comparison of the difference between the contact probability decay for STAG2 WT and KD cells. The average contact probabilities by compartments for STAG2 WT and KD cells were similarly normalized and compared[47]. We first calculated the average Hi-C contacts for pairs of two genomic loci that are labeled by compartment types as A-A, A-B, and B-B,

respectively, and then normalized by the average contacts in the first off-diagonal of Hi-C maps. We did this process for both intra- and inter-chromosomal contacts over all chromosomes.

TADs were identified by using findTADsAndLoops.pl and TADbit[48] in 100 kb bin size We then calculated the TAD length as the difference between the two boundaries for each TAD and plotted the distribution for STAG2 WT and KD cells. A *t*-test on the mean value of the WT/KD distribution was performed to

**Fig. 4 STAG2 regulates the expression of IRF9 and PD-L1 in melanoma via modulating 3D genome organization at the IRF9 locus. a–d** Effects of doxy-inducible STAG2 shRNA knockdown on mRNA (**a**), protein (**b**), and surface expression (**c**, **d**) levels of PD-L1 in human melanoma cells ($n = 3$). **e** STAG2 regulates PD-L1 expression via IRF9 in human M14 cells. **f–h** Effects of CRISPR/Cas9-mediated deletion of the STAG2 binding site at the boundary of the TAD downstream of IRF9 on the expression of IRF9 (**f**) and PD-L1 (**g**, **h**) in human M14 cells ($n = 3$). Three independent clones of binding site deletion or control were used in the analyses. **i** IRF9 enhancer inactivation reversed the regulation of PD-L1 by STAG2 KD in M14 cells. **j–l** Correlation between STAG2 and IRF9 mRNA levels (**j**), IRF9 and PD-L1 mRNA levels (**k**), STAG2 and PD-L1 mRNA levels adjusted for STAT1 (**l**) in TCGA SKCM patients ($n = 473$). All expression values were $\log_2$ transformed. The Spearman's correlation coefficient $R$ value and the two-sided $P$ value are shown. **m** Proportion of responders versus progressors in the overall cohort with high and low STAG2 mRNA expression level (divided by the median). (two-sided Fisher's exact test, $P = 0.04827$). **n** Progression-free survival (PFS) stratified by high versus low *STAG2* mRNA expression level (split by the median) in our overall cohort. Tumors with high *STAG2* had worse PFS (two-sided KM log-rank test, $P = 0.032$). **o** A schematic model depicts the regulation of IRF9 and PD-L1 expression by STAG2 via modulating 3D genome organization at the IRF9 locus. Data in **b**, **c**, **e**, **i** are representative of three independent experiments with similar results. Data in **a**, **d**, **f–h** are presented as mean ± SEM. $P$ values in **a**, **d** were determined by two-tailed ratio paired $t$-tests. $P$ values in **f–h** were determined by unpaired two-tailed $t$-tests compared with the WT cells. Statistical significance is determined as: ns $P > 0.05$; *$P < 0.05$; **$P < 0.01$; ***$P < 0.001$.

evaluate the significance of the difference. Insulation scores were obtained by using hicFindTADs at different resolutions (100 kb, 250 kb, 500 kb, or 1 Mb as indicated) from the HiCExplorer tools. To further investigate the property of boundaries of TADs identified in 100 kb bin size, we defined boundaries with similar (located within 10 Kb) or distinct (apart from 10 Kb) genomic locations between WT and KD samples as stable boundaries (SB) or variable boundaries (VB), respectively. Furthermore, in order to investigate whether we can find any pattern associated with this boundary expansion/shrinking, we defined that VB start sites are always referred to as the sites in the control (WT), and the related end sites are always in the KD. Normalized ChIP-seq signal (RPM: reads per million) from CTCF and cohesin components (including SMC1A, STAG1, and STAG2) at all boundaries were obtained and plotted using deeptools (version: 3.0.2)[49]. Based on the movement of boundary upon STAG2 knockdown, we further defined expanded TADs which have either boundary extended in KD, and shrinked TADs which have either boundary shrinked in KD. Afterward, $\log_2$(KD/WT) of the normalized ChIP-seq signal of STAG1 or STAG2 is averaged from all variable boundaries and plotted from the start to the end of each variable boundary.

**HiChIP**. HiChIP assays were performed according to the previously published protocol[14] with minor modifications. Briefly, ~10 million cells were crosslinked with 1% formaldehyde for 10 min at room temperature and lysed with Hi-C lysis buffer (10 mM Tris-HCl pH 7.5, 10 mM NaCl, 0.2% NP-40) supplemented with protease inhibitors for 30 min at 4 °C. Chromatin was digested by the MboI restriction enzyme for 2 h at 37 °C. The sticky ends of the digested DNA were filled with dCTP, dGTP, dTTP, and biotin-labeled dATP by DNA Polymerase I, Large Klenow Fragment, and then ligated by T4 DNA ligase for 4 h at room temperature. Chromatin was sonicated in Nuclear Lysis buffer (50 mM Tris-HCl pH 7.5, 10 mM EDTA, 1% SDS), 1:5 diluted in ChIP Dilution buffer (16.7 mM Tris-HCl pH 7.5, 1.2 mM EDTA, 1.1% Triton X-100, 167 mM NaCl, 0.01% SDS), and then processed with regular chromatin immunoprecipitation using the H3K27ac antibody (7.5 ug each sample, Abcam, ab4729). The H3K27ac-associated Biotin-labeled DNA fragments were then enriched by Streptavidin C1 magnetic beads (8 ul each sample) and processed with Illumina Nextera DNA library preparation. The DNA libraries were size-selected with Ampure XP beads (300–700 bp) and sequenced by Illumina HiSeq 4000 at the Genomics Core Facility, Icahn School of Medicine at Mount Sinai. The sequencing reads were aligned to hg19 using the HiC-Pro pipeline[45]. H3K27ac peaks were identified by hichip-peaks from the HiChIP data with an FDR level at 0.001[50]. Afterward, the H3K27ac-associated loops were identified using the FitHiChIP[51]. Differential loops between M14 STAG2 WT and KD cells were then identified by diffloop package in R. Briefly, we first removed self-ligated loops and loops with FDR >0.01. We then further filtered loops that are present strongly (≥5 supporting reads) in one replicate but absent (== 0 supporting reads) in the other replicate. The quickAssoc function based on an over-dispersed Poisson regression was used to access differential features. Differential loops with FDR <1% were further separated as enhanced (stronger In KD cells) and impaired loops (weaker In KD cells) based on the fold change and used for downstream analyses. To investigate the different properties of the anchor regions between enhanced and impaired loops, we defined the degree of switch which is the sum of $\log_2$(KD/WT) for STAG1 and STAG2. The positive degree of the switch indicates that the loss of STAG2 is over-compensated by the gain of STAG1, whereas the negative degree of the switch indicates that the loss of STAG2 is under-compensated by the gain of STAG1. Averaged degree of the switch from all enhanced and impaired loops was plotted along the loops with fixed flanking regions extended from left and right anchors using deeptools as described earlier.

**RNA-seq**. Total RNA was extracted using the RNeasy Mini kit (Qiagen, Hilden, Germany) according to the manufacturer's protocol. Quality Control of total RNA was completed using the Bioanalyzer Eukaryote total RNA 6000 Nano kit (Agilent). The concentration of the total RNA was measured using the Qubit RNA HS Assay

Kit (Invitrogen). Library construction and sequencing were carried out at the Center for Computational and Integrative Biology (CCIB) DNA Core at MGH. Briefly, the library was constructed using the NEBNext PolyA mRNA Magnetic Isolation Module & NEBNext Ultra Directional RNA Library Prep Kit for Illumina (New England BioLabs). RNA input was 100 ng of total RNA completed with 15 cycles of PCR. Quality control of the constructed libraries was done by using the Agilent Bioanalyzer High Sensitivity DNA Reagents and quantified using the KAPA Biosystems Library Quantification Kit (Illumina) and Universal qPCR Mix using the BioRad CFX96 Real-Time System. Libraries were loaded on the HiSeq 2000 as a Single Read 50 run. They were pooled and loaded at a concentration of 16 pM with a spike of 1% PHIX. Sequencing reads were mapped to the UCSC hg19 reference genome using STAR version 2.7.0f[52], and the differential expression analysis was performed with rsem version 1.3.0[53]. Normalized RNA-seq data were analyzed via GSEA 2.0[54]. Enrichment analyses were performed using 1000 gene set permutations, a weighted enrichment statistic, the Signal2Noise ranking metric, and gene set minimum and maximum sizes of 15 and 500, respectively.

**Plasmids**. pTRIPZ inducible human STAG2 shRNA (shSTAG2#60) has been described previously[6]. For shSTAG2#60 resistant STAG2 mutant, the mutated STAG2 allele was generated using PCR mutagenesis and verified by sequencing. pLKO construct containing shRNAs against human STAG1(shSTAG1#197: TRCN0000145197) was purchased from Sigma and described previously[6]. Lenti-CRISPR v2 (#52961), lentiCRISPR v2 hygro (#98291), psPAX2 (#12260), and pCMV-VSV-G (#8454) were purchased from Addgene. LCV2-KRAB-dCas9-BSD was previously described[21].

**Reverse-transcription and real-time PCR**. Total RNA was extracted from cells using the RNeasy Mini kit (Qiagen, Hilden, Germany) according to the manufacturer's protocol, and converted to cDNAs using the RevertAid Reverse-Transcription Kit (Thermo Fisher Scientific, Cambridge, MA). Quantitative PCR was performed using the SYBR Green I Master (Roche) reaction mix on a Light Cycler 480 (Roche). Each sample was tested in triplicate, and results were normalized to the expression of the housekeeping gene GAPDH. Relative gene expression was calculated by the $2^{-\Delta\Delta CT}$ method. The following primers were used in this study: IRF9, 5′-CCATCAAAGCGACAGCACAG-3′ and 5′-GCCCCCT CCTCCTCATTATT-3′; PD-L1, 5′-GGCATTTGCTGAACGGCAT-3′ and 5′-CAA TTAGTGCAGCCAGGT-3′; GAPDH, 5′-CAACGAATTTGGCTACAGCA-3′ and 5′-AGGGGTCTACATGGCAACTG-3′.

**Western blotting**. Cells were lysed in a modified lysis buffer (50 mM Tris pH 7.4, 150 mM NaCl, 1% NP-40, 1 mM EDTA, 50 mM NaF, 10 mM β-glycerophosphate, 10 nM calyculin A, 1 mM Na3VO4, and protease inhibitors), followed by SDS-PAGE and western blotting analysis as previously described[55]. The primary antibodies against STAG2 (Santa Cruz, SC-81852), IRF9 (Cell Signaling, #76684), IRF7 (Cell Signaling, #4920), USP18 (Cell Signaling, #4813), ISG15 (Santa Cruz, SC-166755), IRF1 (Cell Signaling, #8478), IRF3 (Cell Signaling, #11904), GAPDH (Cell Signaling, #2118), GAPDH (Cell Signaling, #51332), STAG1 (Novus Biologicals, NB100-298), PD-L1 (R&D Systems, AF156), CTCF (Cell Signaling, #2899), and Flag M2 (Sigma, F3165) were used. All primary antibodies were used at a 1:1,000 dilution, with the exception of GAPDH (1:3000) and PD-L1 (1:200).

**Flow cytometry**. Cells were collected, washed, and resuspended in PBS containing 10% FBS. Single-cell suspensions were stained at a 1:20 dilution with APC anti-human PD-L1 antibody (Biolegend, 329708) or isotype control (Biolegend, 400319) for 30 min on ice. Cells were then washed and resuspended in a 7-AAD viability staining solution (Biolegend, 420404). Dead cells and doublets were excluded on the basis of forward and side scatter. Cells were acquired on a Canto Flow Cytometer (BD FACSCanto, BD Biosciences) using BD FACSDiva software (BD

Biosciences, version 8.0.1). Data were collected for 20,000 cells and analyzed with FlowJo software (FlowJo, Version 10.4).

**CRISPR/Cas9-mediated gene knockout**. Guide RNA (gRNA) targeting IRF9 (GAACTGTGCTGTCGCTTTGA) was designed on the website http://crispor.tefor. net/. gRNA targeting human rosa26 (AGGGCCGCACCCTTCTCCGG) was used as a control. Sense and antisense oligonucleotides were annealed and then cloned into lentiCRISPR v2 hygro (Addgene #98291) according to the "Target Guide Sequence Cloning Protocol" Provided by the laboratory of Dr. Feng Zhang[56].

**CRISPR/Cas9-mediated enhancer repression**. Guide RNAs against putative IRF9 enhancer were identified using the sgRNA designer tool from the Broad Institute[21]. The sequences are:
    IRF9_e1a (ATCGGCCCCCAGAATTCTGG) and IRF9_e1b (GAGTGGAGA AGCTGACCGT). gRNA targeting human rosa26 was used as a control. Oligos were annealed and cloned into the lentiviral vector LCV2-KRAB-dCas9-BSD which encodes a dead Cas9 (dCas9) fused to the transcriptional repressor KRAB.

**CRISPR/Cas9-mediated STAG2 binding site deletion**. Guide RNAs (gRNAs) targeting ~200 bp upstream and downstream sequence of STAG2/CTCF motif were designed using http://crispor.tefor.net/. For Site A, the two guide RNAs were: ACTGCCACCTCCCGTCTACT and GAAGCCCCGTCACTCCCGTG. For Site B, the two guide RNAs were: TTGGGTTCTACACTGTCAGG and CAGGCTGA TGACGCCCCGAG. The potential deleted region was designed to be in the non-coding region (intron). gRNAs were cloned into the lentiCRISPR v2 backbone to generate multiplexed (2xsgRNA) lentiCRISPR_V2 by PCR cloning. As previously descried[21], the sequence "U6 promoter-sgRNA1-scaffold-H1 promoter-sgRNA2" was introduced into the lentiCRISPR v2 backbone by two PCR reactions and a three-fragment ligation. After viral infection, single-cell clones of M14 were picked by sorting 7-AAD negative cells into a 96-well plate using a SONY SH800S cell sorter. Cell clones were screened for deletion of STAG2 binding sites by Sanger sequencing.

**The Cancer Genome Atlas (TCGA) skin cutaneous melanoma (SKCM) dataset analysis**. SKCM RNA-seq data were downloaded from FireBrowse, a companion portal to the Broad Institute GDAC Firehose analysis for curating data generated by TCGA (http://firebrowse.org/, accessed 7/6/2020). The RNA-seq data were normalized with TPM (transcripts per million), containing mRNA expression values of 20,531 genes from 473 samples. Twenty-nine genes with unidentified names were removed from the dataset. The resulting dataset with 20,502 genes was then log2-transformed for downstream correlation analysis. Correlations of mRNA expression levels between STAG2 and adjusted CD274, which encodes PD-L1, were computed with nonparametric Spearman's rank correlation coefficient after adjustment for STAT1 expression based on the predicted effects of STAT1 on CD274 in a univariate linear regression.

**PD-1 immune checkpoint blockade treated melanoma cohort analysis**. Patients in the Schadendorf cohort had advanced melanoma and had received PD-1 immune checkpoint blockade (ICB)[23]. Out of 144 patients, 121 had RNA-seq data available. Based on best objective response (BOR) to anti-PD-1 ICB, 56 patients with the progressed disease (PD) as the best response to therapy were identified as progressors, 47 patients with complete response (CR), or partial response (PR) were identified as responders. Remaining patients with a mixed response (MR) or stable disease (SD) were excluded from downstream analysis. The proportion of responders versus progressors in the 103 included patients with high and low STAG2 mRNA expression level (divided by the median) was calculated. Survival analysis was performed utilizing the R packages *survminer* and *survival*. For Kaplan–Meier curve survival analysis, a two-sided log-rank test was used to compare progression-free survival (PFS) curves. The distribution of STAG2 mRNA expression level was visualized using the R *ggplot2* package. A nonparametric Mann–Whitney–Wilcoxon (MWW) test was used to compare STAG2 mRNA expression levels between responders and progressors. A Fisher's exact test was utilized to examine the association between high and low STAG2 expression (median split) and response to therapy (progressor vs responder). All tests were two-sided unless indicated otherwise.

**Statistics and reproducibility**. All the statistical analyses were performed using GraphPad Prism 8.0 software unless otherwise noted, and error bars indicate SEM or SD. The number of independent experiments, the number of events, and information about the statistical details and methods are indicated in the relevant figure legends. *P* values of less than 0.05 were considered significant. Unless stated otherwise, the experiments were not randomized and investigators were not blinded to allocation during experiments and outcome assessment.

**Reporting summary**. Further information on research design is available in the Nature Research Reporting Summary linked to this article.

## Data availability
The data that support this study are available from the corresponding authors upon reasonable request. High-throughput sequencing data have been deposited to the Gene Expression Omnibus (GEO) data repository with the accession number GSE156773. Source data are provided with this paper.

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

## Acknowledgements

We thank Zhen Chen, Julia Donovan, Lajos Kemeny, Yan Kong, Sebastian Trousil, Jiayi Yu, Ping Yuan, and members of the Zheng Lab for assistance in data collection, analysis, and/or helpful discussion on the manuscript. This work is supported by the Melanoma Research Alliance, Sun Pharma-SID Mid-Career Investigator Award, and funds from MGH (to B.Z.). Lei Gu was supported by the Deutsche Forschungsgemeinschaft (DFG, German Research Foundation), EXC 2026, Cardio-Pulmonary Institute (CPI), Project ID 390649896. We thank Bioinformatics Core Unit (BCU) and Mario Looso, funded by Max Planck Society and Excellence Cluster CPI (Exc 2026), for the assistance with bioinformatics analysis. Lei Gu and Yang Shi were supported by the NIH Outstanding Investigator Award to Y.S. (1R35CA210104) and funds from BCH. Y.S. is an American Cancer Society Research Professor.

## Author contributions

Z.C., L.G., Y.H., X.Z., M.L., S.G., K.G., Y.S., and B.Z. designed experiments; Z.C., Y.H., X.Z., M.L., D.T., C-H.S., and T.Y. performed experiments; L.G., M.H., L.C., Y.Q., Z.N., and A.F. performed the computational analysis; J.C. and D.L. performed TCGA dataset correlational analysis and anti-PD-1 therapy response cohort analysis; D.T.F., E.S., A.P., R.J.S., K.T.F., and G.M.B. provided patient data; all of the authors interpreted data and discussed results; and B.Z. and L.G wrote the paper with input from all of the authors.

## Competing interests

Y.S. is a co-founder and equity holder of Constellation Pharmaceuticals and Athelas Therapeutics, a consultant for Active Motif, and an equity holder of Imago Biosciences. K.T.F has served on the Board of Directors of Clovis Oncology, Strata Oncology, Loxo Oncology, and Checkmate Pharmaceuticals; Scientific Advisory Boards of X4 Pharmaceuticals, PIC Therapeutics, Sanofi, Amgen, Asana, Adaptimmune, Fount, Aeglea, Shattuck Labs, Tolero, Apricity, Oncoceutics, Fog Pharma, Neon, Tvardi, xCures, Monopteros, and Vibliome; consultant to Lilly, Novartis, Genentech, BMS, Merck, Takeda, Verastem, Boston Biomedical, Pierre Fabre, and Debiopharm; and research funding from Novartis and Sanofi. G.M.B has sponsored research agreements with Olink Proteomics, Palleon Therapeutics, InterVenn Biosciences, and Takeda Oncology; served on scientific advisory boards for Novartis, Merck, Nektar Therapeutics, Iovance, and Ankyra Therapeutics; consultant to InterVenn Biosciences, Ankyra Therapeutics, and Merck. The remaining authors declare no competing interests.
