## [Peer Review File · Nature Communications]

REVIEWER COMMENTS

Reviewer #1 (Remarks to the Author):

The authors characterize the effect of STAG2 depletion in a melanoma cell line using a variety of genomics technologies. They identify a role for STAG2 depletion in expanding TADs and enhancing loop formation at sites bound by STAG1. They also identify IRF9 as a target of STAG2. These results are of potential interest to the broader scientific community, since STAG2 is commonly mutated in many different cancer types.

Major Concerns

1. A major concern is the lack of replication of the presented experiments. It appears that only a single replicate was performed for the HiC and ChIP-seq experiments in wild-type (WT) and knockdown (KD) conditions. It is not possible to confirm the number of experiments for the HiC data, since the HiC data and/or results were not included in the GEO submission. However, this appears to be the case from the Supplementary Table. In addition, only a single cell line and a single shRNA was used in the study. However, the shRNA appears to be on target, since a rescue experiment was performed. It appears that the authors have 3 melanoma cell lines that stably express the STAG2 shRNA. Replicating the reported findings across these lines would significantly strengthen their findings.
2. The depletion of STAG2 binding to the chromatin under knockdown appears modest and it appears that STAG2 is still present at many of its original binding sites. The authors utilize a log₂ ratio of STAG2 WT/KD to illustrate the effect of the KD experiment. It appears that the KD only results in about a 20% reduction in signal intensity across sites. What is the level of KD observed for STAG2 at the protein level? It does not appear to be quantified in the paper. Although, a single western blot is provided in Figure 4. What is the average decrease in STAG2 signal at STAG2 binding sites? The paper would benefit from showing example ChIP-Seq signal tracks for the WT and KD experiments. Only the called peaks were included in the GEO submission, therefore it is not possible to verify the signal intensities. However, the majority of the peaks from the KD condition are called in the wild-type condition.
3. How were the lost/gained STAG2 sites called from the ChIP-seq experiments of the WT and KD? What method was used to determine these sites?
4. Are the subtle changes observed in the HiC data caused by variability in the HiC data? Are they reproducible? Please see point one.
5. The text is missing descriptive statistics of the results throughout. For example, what were the number of variable and stable TAD boundaries observed? What was the p-value for the significant difference in the HiC TAD and HiChIP loop length between WT and KD? Is the gain of H3K27ac at switch sites significant? What is the change in IRF9 expression and its p-value/FDR q-value for RNA-Seq experiment? Where does IRF9 rank in the differentially expressed genes? Was the increased expression of the other members of the IFN pathway significant? Etc...
7. Were the rescue experiments and over-expression experiments completed in triplicate? Are they significant?
8. Of interest to the other reported findings, can the authors rescue the observed changes in the chromatin interaction data at the IRF9 locus? Or in general?

8. A parametric T-test was used to calculate significance for comparisons between WT and KD conditions. For example, the box-plots of the TAD length reveal highly skewed non-normal distributions. A Non-parametric test may be more appropriate for these analyses.

Reviewer #2 (Remarks to the Author):

Chu et al. perform knockdown of the cohesin subunit STAG2 in melanoma cell lines, and characterise the effects of the knockdown using a comprehensive set of genomic techniques including ChIP-seq, RNA-seq, Hi-C and HiChIP. They find that STAG2 knockdown is compensated by STAG1 at a subset of sites, and leads to changes in TADs and H3K27ac-associated loops. They also analyse the role and regulation of IRF9. The function of IRF9 is outside my area of expertise so I have not focused on this part of the manuscript in detail. Overall, the analysis is well-structured, the results are interesting, and the data shown supports the conclusions. However, some details that I would expect to see are missing and, in some places, additional information would be helpful to better understand the nature of the STAG2-dependent changes.

Major points

- It would be helpful to include a more thorough characterisation of the effects of the STAG2 knockdown, for context. The knockdown is clearly not complete as there is still ChIP-seq signal for STAG2 – what is the level of knockdown at the mRNA and protein levels? Are there effects on the viability or phenotype of the cells? Even if this system has been used in previous publications, readers will appreciate a brief summary here.**
- Related to this, STAG non-switch peaks clearly have lower signal overall in both WT and KD as well as lower CTCF and SMC1A signal as mentioned. Do they have other differences e.g. position relative to domains/boundaries vs enhancers/promoters?**
- The IRF9 CRISPR KO is also not characterised. My understanding is that no selection is performed, so this is likely to be a heterogenous population. While the effect is clear, the level of KO should be quantified to provide context for this change.**
- One interesting and in my opinion under-explored finding here is the change in TAD size in the STAG2 KD. The analysis largely focuses on comparing overall TAD size and boundary positions, with few examples shown. This left me with questions about the nature of the changes in TADs and boundaries. Are completely new boundaries being formed, or are neighbouring TADs merging to form larger TADs while sub-domain boundaries become domain boundaries? It would be helpful to show more examples of the actual Hi-C data to illustrate this e.g. in Fig 3d.**
- In addition, it is not ideal to compare TADs by only comparing boundary positions, as TAD and boundary calls are binary and small changes between samples due to noise or sequencing depth can lead to a region falling just below a threshold rather than just above it, for example. In particular, since variable boundaries have lower levels of CTCF, and cohesin binding in general, some of these could be borderline boundary calls that vary by chance. It would be informative to know whether boundaries are truly lost/gained or this is really a change in strength. This could be addressed by calculating insulation scores (as in Crane et al. 2015, PMID: 26030525) to quantitatively measure boundary strength. This would help better understand the role of STAG1/2 at boundaries.**

Minor points

- The authors should check their phrasing choices for accuracy, as I found some of these confusing: e.g. in Fig 1f “difference in TAD boundary’s length” should presumably be “difference in TAD size” or similar, since as far as I understand this figure refers to TADs**

not boundaries.

- Fig 1h is not clear to me – is this at the boundary of the TADs in the control or the shifted boundary in the knockdown?
- Fig 2b, most changing loops are within TADs: this is presumably also true of most loops in general – are the proportions significantly different?
- Fig 2d, super-enhancers: because of the way super-enhancer calls work, a small change in H3K27ac signal can change whether a region is identified as a super enhancer or not. Quantitative comparisons e.g. a heatmap or metaplot of H3K27ac across SEs would reveal whether this is really a relevant level of change.
- The region shown in Fig 3d is very large, so it is not clear where exactly the loop anchors are. Both loop anchors seem to be close to a promoter and a strong CTCF peak. Is this a promoter-promoter / CTCF-CTCF loop as well as involving an enhancer?
- What are the horizontal dashed lines in 3d?
- WT/KD and KD/WT are both used in different places. This is confusing.
-

Reviewer #3 (Remarks to the Author):

STAG2 is frequently mutated in several human cancers. STAG2 is a subunit of cohesin complexes, which have important roles in forming chromatin loops, TADs and enhancer-promoter interactions. Somatic cells express cohesin complexes that contain either STAG2 or its paralog STAG1. Loss of both STAG1 and STAG2 causes cell lethality, but cohesin complexes containing STAG1 can compensate for the loss of STAG2 to the extent that cells remain viable. Loss of STAG2 is known to increase the chromatin association of STAG1, to change its genomic distribution, to alter genome architecture, in part because complexes containing STAG1 have a longer residence time on chromatin and thus form longer chromatin loops, and STAG2 is thought to have a tumor suppressor function by affecting gene regulation through its genome architecture function. However, which genes are mis-regulated upon loss of STAG2 and how this contributes to tumorigenesis in which cancer type is poorly understood.

In this manuscript Chu et al. report the results of experiments in which they have analyzed the consequences of partial STAG2 loss, experimentally created with an inducible shRNA, in a melanoma cell line (M14) on chromatin structure, H3K27ac levels and gene expression. First, the authors describe changes in cohesin and CTCF occupancy by using ChIP-seq after STAG2 depletion. Upon STAG2 depletion, some STAG2 sites were occupied by STAG1 whereas others had reduced STAG2 levels. The authors refer to these sites as 'STAG switch sites' and 'STAG non-switch sites', respectively. As expected, CTCF occupancy did not change upon STAG2 depletion. In terms of chromatin organization, compartmentalization was slightly reduced in STAG2 depleted cells, whereas some TADs increased in size or shrunk. The authors refer to these as 'expanded TADs' and 'shrunk TADs', respectively. At the borders of expanded TADs an increase in STAG1 binding was detected. These confirm and extend previous reports about the effects of STAG2 loss.

Next, the authors performed HiChIP experiments using antibodies to H3K27ac, an enhancer mark to identify potential changes in promoter-enhancer looping in STAG2 depleted cells. These experiments identified numerous loops, which were enhanced in abundance upon STAG2 loss, and others, which became less abundant (in both cases about 10,000 loops were identified). For the loops, which increased in abundance, increased levels of H3K27Ac were detected.

By combining H3K27Ac HiChIP, RNA seq and STAG2 ChIP data obtained from cells, from which STAG2 was depleted or not, the authors identified 148 mis-regulated genes that lose STAG2 binding and coincide with changes in H3K27Ac interactions. In subsequent

experiments the authors focused on IRF9, a transcription factor important for the type I interferon signaling pathway. The authors found that STAG2 depletion leads to upregulation of IRF9 in M14 and two other melanoma cell lines (Colo 829 and Mel 1617). Furthermore, the authors report that in M14 cells, STAG2 depletion increased the size of a TAD adjacent to the IRF9 gene and led to a new interaction between the IRF9 promoter and a distant enhancer, as detected by H3K27HiChIP.

Finally, the authors showed that STAG2 depletion caused an increase in PD-L1 protein levels and provide evidence that this phenomenon depends on IRF9. They found that deletion of a cohesin binding site between the IRF9 gene and the adjacent TAD also led to upregulation of IRF9 and PD-L1, even without STAG2 depletion, whereas deletion of another cohesin binding site did not have these effects. Moreover, the authors applied the KRAB-dCas9 system to repress the putative distal enhancer of IRF9 identified in H3K27ac HiChIP analysis and found that under these conditions STAG2 depletion did not cause upregulation of IRF9 and PD-L1. Finally, the authors analyzed RNA-seq data in the cancer genome atlas and found correlations between the expression levels of STAG2, IRF9 and PD-L1 that are consistent with their findings.

How STAG2 loss alters the genomic distribution of cohesin, genome architecture and gene regulation has been studied in some detail. But the finding that the changes in genome architecture that are caused by STAG2 loss lead to upregulation of IRF9 in melanoma cells, and that this in turn increases PD-L1 expression, is novel and potentially interesting. I would therefore support publication of this manuscript, provided that the points below will be addressed.

1. The authors claim that their findings 'reveal a previously unappreciated role of STAG2 in genome architecture ...' (page 9), but this is an overstatement since previous studies already reported that in the absence of STAG2 (a) STAG1 levels are increased on chromatin, (b) STAG1 accumulates at sites that are normally occupied by STAG2, and (c) that chromatin loops become longer (Kojic et al., 2018; Wutz et al., 2020). These previous results need to be discussed and the authors should focus their manuscript on the novel findings from their experiments.

2. Similarly, previous studies, which have analyzed the effects of cohesin depletion on enhancer-promoter interactions should be mentioned: Guo, Monahan et al., 2012; El Khattabi et al., 2019; Thiecke et al., 2020, and instead of citing several reviews it would be better to cite the original studies, which showed that cohesin is required for chromatin looping and TADs (Gassler et al., 2017; Rao et al., 2017; Schwarzer et al., 2017; Wutz et al., 2017).

3. In Figure 3d, it would be important to show the original Hi-C data, in which an increase in the TAD next to IRF9 can be seen and not just a schematic drawing.

4. In the same figure panel, it is unclear at which sites STAG2 levels are decreased upon induction of shRNA targeting STAG2 expression. Some sites seem unaffected, some reduced but some actually increased. This needs to be explained.

5. Extended Data Figure 3b. A P value is mentioned in the figure legend but not shown in the figure.

6. Page 7, lines 19-21. "We found that knockdown of STAG2 increased both mRNA and protein levels, as well as surface expression of PD-L1 in melanoma cells (Fig. 4a-d, Extended Data Fig. 5c-g)." It seems that this is shown in Extended Data Fig. 5e-g, not in Extended Data Fig. 5c and d.

Point-by-point Response Letter

We would like to thank all three reviewers for their insightful and constructive comments. We have carefully addressed all comments, added new experimental data and revised the manuscript accordingly. We are confident that the manuscript has now been strengthened with the revision, and hope it will be considered acceptable for publication.

Reviewer #1

1. A major concern is the lack of replication of the presented experiments. It appears that only a single replicate was performed for the HiC and ChIP-seq experiments in wild-type (WT) and knockdown (KD) conditions. It is not possible to confirm the number of experiments for the HiC data, since the HiC data and/or results were not included in the GEO submission. However, this appears to be the case from the Supplementary Table. In addition, only a single cell line and a single shRNA was used in the study. However, the shRNA appears to be on target, since a rescue experiment was performed. It appears that the authors have 3 melanoma cell lines that stably express the STAG2 shRNA. Replicating the reported findings across these lines would significantly strengthen their findings.

Response: We thank the reviewer for this important comment. We have now included ChIP-seq data of STAG1/2, SMC1A and CTCF from three biological replicates in our analyses (Revised **Fig. 1a**, **Supplementary Fig. 1**). Indeed, we have only used a single melanoma cell line, M14, for the Hi-C study. Upon identification of IRF9 as a candidate target of STAG2 from M14 cells, we have confirmed the regulation of IRF9 expression by STAG2 in multiple cell lines, including shRNA knockdown of STAG2 in Colo829 and Mel1617 cells and ectopic expression of STAG2 in WM902BR cells (**Fig. 3e and Fig. S7i-j**). Unfortunately, due to the limited resource, at this point we could not extend our Hi-C study to other melanoma cell lines, but have now pointed out this potential limitation in the discussion section. We have also acknowledged the limitation of drawing conclusion from single replicate of Hi-C experiment and tuned down conclusion related to our Hi-C study. We hope the reviewer would put more weight on other strengths and novel findings of our manuscript, such as H3K27Ac HiChIP analysis (two biological replicates were included), identification of a novel target of STAG2 in multiple melanoma cell lines, and the implication of STAG2 in regulating PD-L1 expression and sensitivity to anti-PD-1 therapy in cancer. With the rapid advancement of various chromatin conformation capture techniques at higher resolution, we anticipate we will be able to gain more insight into the role of STAG2 in the 3D genome organization in future studies.

2. The depletion of STAG2 binding to the chromatin under knockdown appears modest and it appears that STAG2 is still present at many of its original binding sites. The authors utilize a log₂ ratio of STAG2 WT/KD to illustrate the effect of the KD experiment. It appears that the KD only results in about a 20% reduction in signal intensity across sites. What is the level of KD observed for STAG2 at the protein level? It does not appear to be quantified in the paper. Although, a single western blot is provided in Figure 4. What is the average decrease in STAG2 signal at STAG2 binding sites? The paper would benefit

from showing example ChIP-Seq signal tracks for the WT and KD experiments. Only the called peaks were included in the GEO submission, therefore it is not possible to verify the signal intensities. However, the majority of the peaks from the KD condition are called in the wild-type condition.

Response: Based on semi-quantification analyses of Western blot results, the level of STAG2 KD is about 85% in our inducible knockdown melanoma cells. We have now included data from the semi-quantification analyses in **Supplementary Fig. 7k-l**. We have also added three examples of ChIP-seq signal tracks for loss of STAG2 binding sites (**Supplementary Fig. 1b**).

Regarding the discrepancy between the levels of STAG2 protein KD and reduction of STAG2 binding intensity, we noted that only 11% of all STAG2 sites have their binding intensities reduced for more than 50% upon STAG2 knockdown, suggesting that the effect of STAG2 KD on its chromatin binding is site-specific. Because the standard ChIP-seq analysis normalize the signal based on the total number of reads, it has been challenging to measure globally increased or decreased ChIP-seq signal. Therefore, it is possible that knockdown of STAG2 has more remarkable global effects on STAG2 binding than observed from our current ChIP-seq results. Although the spike-in approach would be ideal for the comparative ChIP-seq data analysis of the same protein with different abundance (WT vs. KD), it is impossible in our case as the STAG2 antibody used in our ChIP-seq recognizes only human and mouse STAG2 proteins. Lastly, regarding peak calling, we have now uploaded the bigwig files to GEO so that the signal intensities can be verified.

3. How were the lost/gained STAG2 sites called from the ChIP-seq experiments of the WT and KD? What method was used to determine these sites?

Response: We have added the following information regarding how differential STAG2 binding site calling in the method section. In brief, we treated WT and KD experiments as 'treatment' and 'control' using the macs2 callpeak function, or *vice versa*, which yielded peaks that are specific to WT or KD.

The differential peak of STAG2 including lost/gained are called by macs2 with the following commands:

- `macs2 callpeak -t old_SA2_sa2kd.bam new_SA2_sa2kd.bam -c old_SA2_sa2wt.bam new_SA2_sa2wt.bam -n SA2_kd2wt -g hs -p 0.01 --shift -75 --extsize 150 --nomodel`
- `macs2 callpeak -t old_SA2_sa2wt.bam new_SA2_sa2wt.bam -c old_SA2_sa2kd.bam new_SA2_sa2kd.bam -n SA2_wt2kd -g hs -p 0.01 --shift -75 --extsize 150 --nomodel`

4. Are the subtle changes observed in the HiC data caused by variability in the HiC data? Are they reproducible? Please see point one.

Response: We thank the reviewer for this important comment. Indeed, with our Hi-C dataset, we cannot rule out the possibility that a mix of technical and biological variations could contribute to the changes we observed in our Hi-C experiment. As a way to mitigate this limitation, we integrated other types of data, such as HiChIP data with biological replicates available, and validated key findings on the IRF9 locus using the CRISPR-based STAG2 binding site deletion and the KRAB-dCas9 enhancer repression approaches, thus preventing artifacts. We have acknowledged the limitation of drawing conclusion from single replicate of Hi-C experiment and tuned down conclusion related to our Hi-C study. Although there could be some false positive changes caused by variability in Hi-C data, we believe our key conclusion on IRF9 as a target of STAG2 has been verified by subsequent functional characterization and wet lab experiments. Please also refer to our response to the first comment.

5. The text is missing descriptive statistics of the results throughout. For example, what were the number of variable and stable TAD boundaries observed? What was the p-value for the significant difference in the HiC TAD and HiChIP loop length between WT and KD? Is the gain of H3K27ac at switch sites significant? What is the change in IRF9 expression and its p-value/FDR q-value for RNA-Seq experiment? Where does IRF9 rank in the differentially expressed genes? Was the increased expression of the other members of the IFN pathway significant? Etc...

Response: We apologize that we did not include some detailed methodology information due to space/word limitation in our original manuscript, which was previously submitted to *Nature* and then transferred to *Nature Communications*. We have now added those statistics into the main text. Specific responses to the questions raised are shown below.

- For example, what were the number of variable and stable TAD boundaries observed?
There were 3789 and 3496 TADs identified from WT and KD, respectively. We then expanded a given TAD's left and right genomic coordinates with 10kb on both directions and defined them as its two boundary regions for further downstream analysis. If two boundary regions were overlapped, they were merged as a single larger boundary. In total, there were 6260 and 5761 boundaries identified from WT and KD, respectively. We then determined the overlap of the boundaries between WT and KD, and separated them as 4380 stable boundary (overlapped) and 3281 variable boundary (non-overlapped).
- What was the p-value for the significant difference in the HiC TAD and HiChIP loop length between WT and KD?
We re-calculated the *p* value by using the Mann-Whitney U Test. The updated *p* values are 3.488e-13 and < 2.2e-16 for the significance of the difference for HiC TAD and HiChIP loop length, respectively, between WT and KD. We added the *p* values in the revised main text.
- Is the gain of H3K27ac at switch sites significant?

We extracted average H3K27ac signal for each switched and non-switched STAG2 peaks. The signal significantly increased at the switch sites ($p < 2.2e-16$, wilcox test), but not at the non-switch sites ($p = 0.1368$, wilcox test).

- **What is the change in IRF9 expression and its p-value/FDR q-value for RNA-Seq experiment?**
Level of IRF9 is 6 times higher in STAG2 KD cells than in WD cells, and the FDR q value is 0.
- **Where does IRF9 rank in the differentially expressed genes?**
Based on the ranking of fold change, IRF9 was at the 44th position of the significantly upregulated gene list.
- **Was the increased expression of the other members of the IFN pathway significant?**
Our gene set enrichment analysis revealed IFN α response and IFN γ response as the top two most significantly enriched Hallmark pathways induced by STAG2 knockdown (**Fig. 3b and Supplementary Fig. 7a, b**). Increased expression levels of several IFN pathway members were found to be significant. For examples, IRF7 (q=0, fc=10.8), USP18 (q=0, fc=20.8), ISG15 (q=0, fc=51.8). Their protein levels were also confirmed to increase in KD cells (**Fig. 3f**).

6. Were the rescue experiments and over-expression experiments completed in triplicate? Are they significant?

Response: Yes, the rescue experiments and over-expression experiments were completed in triplicates, and significant differences were observed. We have now specified the number of replicates in the figure legends, carried out additional quantification analyses and included results from the statistical analysis in the figures (**Supplementary Fig. 7e-g, Supplementary Fig. 7i-j**).

7. Of interest to the other reported findings, can the authors rescue the observed changes in the chromatin interaction data at the IRF9 locus? Or in general?

Response: We thank the reviewer for this important comment. In our original manuscript, we showed that knockdown of STAG2 by inducible shRNA increased the expression of *IRF9* in various melanoma cells. Importantly, we further showed that expression of an shRNA-refractory mutant of STAG2 rescued the effects of STAG2 KD on *IRF9* expression in M14 cells, demonstrating that *IRF9* is a direct target of STAG2. Unfortunately, due to the limited resource, we are unable to carry out Hi-C in these cells with rescue at this point. We hope our functional characterization experiments in multiple cell lines would convince readers the critical role of STAG2 in regulating *IRF9* expression in melanoma.

8. A parametric T-test was used to calculate significance for comparisons between WT and KD conditions. For example, the box-plots of the TAD length reveal highly skewed non-normal distributions. A Non-parametric test may be more appropriate for these analyses.

Response: We thank the reviewer for the suggestion. We have now carried out statistical analyses using wilcox test, a non-parametric test, and revised the results in Figures and text.

Reviewer #2

Major points

1. It would be helpful to include a more thorough characterisation of the effects of the STAG2 knockdown, for context. The knockdown is clearly not complete as there is still ChIP-seq signal for STAG2 – what is the level of knockdown at the mRNA and protein levels? Are there effects on the viability or phenotype of the cells? Even if this system has been used in previous publications, readers will appreciate a brief summary here.

Response: Thank you for this comment. Based on RT-qPCR analysis and semi-quantification analyses of Western blot results, we estimate the level of STAG2 KD is ~ 87% in mRNA and ~ 85 % in protein in our inducible knockdown melanoma cells (**Supplementary Fig. 7k-l**). We have previously shown that, although knockdown of STAG2 in BRAF mutant melanoma cell lines did not significantly induces changes in cell cycle progression and apoptosis under normal growth condition (in the absence of BRAF inhibitors), it markedly impaired the response of melanoma cells to BRAF inhibitors. We have now included a brief summary in the introduction section.

2. Related to this, STAG non-switch peaks clearly have lower signal overall in both WT and KD as well as lower CTCF and SMC1A signal as mentioned. Do they have other differences e.g. position relative to domains/boundaries vs enhancers/promoters?

Response: Thank you very much for the suggestion. We have analyzed the STAG non-switch and switch peaks in the context of domain/boundaries and enhancers/promoters, and found that the switch/non-switch pattern was significantly associated with TAD domain boundaries and enhancers, but not promoters. These results are summarized in **Supplementary table 2**.

3. The IRF9 CRISPR KO is also not characterised. My understanding is that no selection is performed, so this is likely to be a heterogenous population. While the effect is clear, the level of KO should be quantified to provide context for this change.

Response: Thank you for this comment. Based on RT-qPCR analysis and semi-quantification analyses of Western blot results, we estimate the level of IRF9 KD is ~ 90% in protein in our inducible knockdown melanoma cells (**Supplementary Fig. 7m**).

4. One interesting and in my opinion under-explored finding here is the change in TAD size in the STAG2 KD. The analysis largely focuses on comparing overall TAD size and boundary positions, with few examples shown. This left me with questions about the nature of the changes in TADs and boundaries. Are completely new boundaries being formed, or are neighboring TADs merging to form larger TADs while sub-domain boundaries become domain boundaries? It would be helpful to show more examples of the actual Hi-C data to illustrate this e.g. in Fig 3d.

Response: Thank you for this important comment. We have carried out further analyses on the changes in TADs upon STAG2 KD in melanoma cells. We observed that 18% of changed TADs had completely new boundaries being formed, 56% of them had neighbouring TADs merged and 26% of them had either boundary shifted (**Supplementary Fig. 5a**), which is probably caused by the setup of isolation strength cut-off. Examples of actual Hi-C data for TAD boundary changes upon STAG2 KD are now included in **Supplementary Fig. 5c-d**.

5. In addition, it is not ideal to compare TADs by only comparing boundary positions, as TAD and boundary calls are binary and small changes between samples due to noise or sequencing depth can lead to a region falling just below a threshold rather than just above it, for example. In particular, since variable boundaries have lower levels of CTCF, and cohesin binding in general, some of these could be borderline boundary calls that vary by chance. It would be informative to know whether boundaries are truly lost/gained or this is really a change in strength. This could be addressed by calculating insulation scores (as in Crane et al. 2015, PMID: 26030525) to quantitatively measure boundary strength. This would help better understand the role of STAG1/2 at boundaries.

Response: Thank you for this important comment and insight. We have compared the insulation profile between variable and stable boundaries, and found that the insulation levels at stable boundaries are significantly higher than those at variable boundaries (**Supplementary Fig 5b**). Indeed, as pointed out by the reviewer, the boundary is usually not truly lost/gained in a population of cells, but rather the strength of boundary changes. We have also revised the text in manuscript to describe these observations.

Minor points

1. The authors should be check their phrasing choices for accuracy, as I found some of these confusing: e.g. in Fig 1f “difference in TAD boundary’s length” should presumably be “difference in TAD size” or similar, since as far as I understand this figure refers to TADs not boundaries.

Response: Thank you for this correction. We have revised the text as suggested.

2. Fig 1h is not clear to me – is this at the boundary of the TADs in the control or the shifted boundary in the knockdown?

Response: We apologize for the confusion. As shown in the figure below, variable boundary (VB) start site is always referred to the site in the control (WT), and the related end site is always in the KD. We have now included these information in the method section. Our intention was to investigate whether we can find any pattern associated with this boundary expansion/shrinking.

3. Fig 2b, most changing loops are within TADs: this is presumably also true of most loops in general – are the proportions significantly different?

Response: We did not observe significant differences between enhanced and impaired loops in terms of within versus across TADs.

4. Fig 2d, super-enhancers: because of the way super-enhancer calls work, a small change in H3K27ac signal can change whether a region is identified as a super enhancer or not. Quantitative comparisons e.g. a heatmap or metaplot of H3K27ac across SEs would reveal whether this is really a relevant level of change.

Response: Thank you for this comment. We have carried out quantitative comparisons for 3 types of SEs, and generated metaplots of H3K27ac across SEs (**Supplementary Fig. 6g**). Indeed, as the reviewer pointed out, the gain and loss of SEs upon STAG2 knockdown corresponding changes in H3K27ac levels.

5. The region shown in Fig 3d is very large, so it is not clear where exactly the loop anchors are. Both loop anchors seem to be close to a promoter and a strong CTCF peak. Is this a promoter-promoter / CTCF-CTCF loop as well as involving an enhancer?

Response: Thank you for the comment. As shown in the new **Supplementary Fig. 8a-b**, our data indicate that the CTCF peak in the IRF9 locus in Fig. 3d is overlapped with the promoter, so the loop represents a promoter-promoter / CTCF-CTCF loop, and the H3K27ac at the promoter of TINF2 serves as an enhancer for IRF9 via the CTCF-CTCF interaction.

6. What are the horizontal dashed lines in 3d?

Response: Those dash lines were used to help visualizing the differences in peak heights. We have now removed these lines to keep the figure clean.

7. WT/KD and KD/WT are both used in different places. This is confusing.

Response: Thank you for the suggestion. We have revised the text to keep them consistent.

Reviewer #3

1. The authors claim that their findings ‘reveal a previously unappreciated role of STAG2 in genome architecture ...’ (page 9), but this is an overstatement since previous studies already reported that in the absence of STAG2 (a) STAG1 levels are increased on chromatin, (b) STAG1 accumulates at sites that are normally occupied by STAG2, and

(c) that chromatin loops become longer (Kojic et al., 2018; Wutz et al., 2020). These previous results need to be discussed and the authors should focus their manuscript on the novel findings from their experiments.

2. Similarly, previous studies, which have analyzed the effects of cohesin depletion on enhancer-promoter interactions should be mentioned: Guo, Monahan et al., 2012; El Khattabi et al., 2019; Thiecke et al., 2020, and instead of citing several reviews it would be better to cite the original studies, which showed that cohesin is required for chromatin looping and TADs (Gassler et al., 2017; Rao et al., 2017; Schwarzer et al., 2017; Wutz et al., 2017).

Response: Thank you for these comments. We have revised the text to include more discussions on previous studies on STAG1/2 and cohesin as suggested.

3. In Figure 3d, it would be important to show the original Hi-C data, in which an increase in the TAD next to IRF9 can be seen and not just a schematic drawing.

Response: Thank you for the suggestion. Since the resolution of our HiC data is not very high, the 100 kb difference is not clear to visualize by eye. That is why we used a schematic drawing to depict the change of TAD next to IRF9. We have now included the original Hi-C data in the **Supplementary Fig. 8c**. We also added the insulation profile across the region, which would make it easier to recognize the change of boundary (**Supplementary Fig. 8d**).

4. In the same figure panel, it is unclear at which sites STAG2 levels are decreased upon induction of shRNA targeting STAG2 expression. Some sites seem unaffected, some reduced but some actually increased. This needs to be explained.

Response: Thank you for this comment. Indeed, not all STAG2 binding decreased upon knock down of STAG2, as our STAG2 shRNA knockdown efficiency only achieved at around 80% at the protein level. We noted that only 11% of all STAG2 sites have their binding intensities reduced for more than 50% upon STAG2 knockdown, suggesting that the effect of STAG2 KD on its chromatin binding is site-specific. In addition, it is possible that only a fraction of the total STAG2 protein in the cells is bound to chromatin. Knockdown of STAG2 by shRNA may first affect the non-chromatin pool of STAG2 proteins, while any remaining STAG2 proteins in the KD cells are preferentially bound to DNA to maintain genome stability. On the other hand, with the greatly reduced total STAG2 proteins in the KD cells, high-affinity STAG2 binding sites may be able to retain STAG2 protein at the expense of low-affinity sites. Upon regional normalization, some high-affinity STAG2 binding sites that were not affected by knockdown of STAG2 could even show a relatively increased signal.

5. Extended Data Figure 3b. A P value is mentioned in the figure legend but not shown in the figure.

Response: We have now indicated the p value in this Figure, now **Supplementary Fig. 4b**.

6. Page 7, lines 19-21. “We found that knockdown of STAG2 increased both mRNA and protein levels, as well as surface expression of PD-L1 in melanoma cells (Fig. 4a-d, Extended Data Fig. 5c-g).” It seems that this is shown in Extended Data Fig. 5e-g, not in Extended Data Fig. 5c and d.

Response: Thank you for this correction. We have revised the text accordingly.

REVIEWER COMMENTS

Reviewer #1 (Remarks to the Author):

The authors were very responsive in their responses to my comments. However, I still have a few concerns.

1. The impact of the knockdown of STAG2 on the ChIP-seq data (ie. the recruitment of STAG2 to the chromatin) should be reported. How many sites were affected? Gained, lost or unchanged after knockdown? This isn't stated anywhere in the manuscript. In addition, it is difficult to assess in the Circos plot and to reconcile figure 1c with the STAG2 profiles presented in figure 3d. The only data provided is the number of promoters affected in figure 3a. The authors did provide some of this information in the response, but it was not incorporated into the text.
2. In addition to the Western blot quantification of the knockdown efficiency, can the authors also demonstrate the knockdown efficiency using the RNA-Seq data? In addition, is there any evidence of differential isoform expression?
3. Globally the ChIP-seq replicates appear to be very similar, but what about the differential STAG2 sites? Do they replicate?
4. Does the change in intensity of the differential STAG2 sites correlate with changes in gene expression using the RNA-Seq data? Also, a volcano plot of the differential gene expression analysis, with the location of IRF9, would benefit the paper.
5. Do the changes observed regarding the TADs in the HiC data correlate with changes in gene expression from the RNA-Seq data?
6. Since the reported findings regarding TADs from the HiC analysis differ somewhat from published findings. The authors should discuss these results. For example, the results of a recent paper by Richart et al. (2021) NAR, which included two shRNAs against STAG2 and HiC replicates, report within TAD changes in chromatin interactions, but little changes affecting compartments and/or boundaries.
7. The TRC clone ID, TRCN0000152538, referenced by the authors (referred to as shSTAG2#38) in the methods does not target STAG2, but DIS3L. Can the authors clarify if this is or is not the correct shRNA?

Reviewer #2 (Remarks to the Author):

The revised manuscript is much clearer due to the additional detail the authors have added, and I'm satisfied with the authors' responses to most of my previous comments. However, there are a few remaining major issues.

I find the Hi-C analysis somewhat unconvincing and lacking in important methodological details that would allow readers and reviewers to assess it fully. I realise that resources can limit the number of replicates and sequencing depth, however different parameters (i.e. a bin size appropriate for the sequencing depth) or alternative visualisations could clarify the results.

- The Hi-C resolution used for each analysis or visualization should be specified in the text and/or figure legends, as well as in the methods. The methods currently only mention a resolution of 1Mb for compartment analysis.

- The insulation score analysis potentially strengthens the findings, however it's unclear how to interpret this as the methods section for this is missing. Typically a more negative insulation score indicates a stronger boundary, however here the scores are positive. If the authors have multiplied the scores by -1 they should say so.
- I find the examples of TAD boundary loss / gain in Supplementary Fig 5 unclear. It looks like there are changes in interaction strength within TADs in the KD/WT panel, but not any clear examples of boundary loss/gain. Showing the insulation scores or TAD boundary scores alongside the matrices, or highlighting the supposedly changing boundaries with an arrow, might make it clearer. I would also suggest using a larger bin size, as at the current resolution the visualization is quite noisy – a lower resolution might be clearer.
- Supplementary Fig 8 focusing on the IRF9 TAD is also unclear, although here at least the difference in TAD structure is visible. It's difficult to understand the positional relationship between the data shown in different panels as some have genomic coordinates and some have gene tracks, but none have both presented legibly.
- I still think it would be helpful to show some visualization, not just a schematic, of the Hi-C data at this locus in a main panel. Perhaps virtual 4C from the IRF9 promoter would make the difference in boundary position visible? Or, again, a lower resolution might actually make the difference in the Hi-C data clearer.

As noted by another reviewer, the authors should focus on their novel results and tone down the claim (page 9) that the role of STAG2 in 3D genome organization and transcriptional control is novel as this has been previously shown by e.g. Kojic et al 2018, Viny et al. 2019.

Minor comments:

The text "difference in TAD boundary's length" is still in Fig 1f. I think this should be "difference in TAD size", based on the updated main text.

Fig 1h is now clear to me, but only after seeing the schematic included in the response to reviewers. This should be included in the main figure.

At the bottom of page 5, the phrase "the switch/non-switch pattern" is not clear. Is it switch sites or non-switch sites that are significantly associated with TADs, boundaries and enhancers?

While the authors note in the response to reviewers that the fact that the majority of altered loops occur within TADs is not statistically significant, this should also be mentioned in the text, otherwise it sounds like an unexpected finding.

Point-by-point Response Letter

We would like to thank reviewers for their insightful and constructive comments. We have carefully addressed all comments, carried out new analyses and revised the manuscript accordingly. We are confident that the manuscript has now been strengthened with the revision, and hope it will be considered acceptable for publication.

Reviewer #1

1. The impact of the knockdown of STAG2 on the ChIP-seq data (ie. the recruitment of STAG2 to the chromatin) should be reported. How many sites were affected? Gained, lost or unchanged after knockdown? This isn't stated anywhere in the manuscript. In addition, it is difficult to assess in the Circos plot and to reconcile figure 1c with the STAG2 profiles presented in figure 3d. The only data provided is the number of promoters affected in figure 3a. The authors did provide some of this information in the response, but it was not incorporated in to the text.

Response: In total, 79,115 peaks were identified in STAG2 WT samples. Upon STAG2 KD, we identified 32,647 peaks with significant loss of binding activity, and 998 with gain of binding activity. As for STAG1 binding profile, there were 285 peaks showing significant loss of binding activity and 30,621 peaks with significant gain of binding activity upon STAG2 KD. As expected, the genomic distributions of differential peaks between STAG1 and STAG2 were very similar. The majority of the differential peaks were located in Introns and Intergenic regions (**Supplementary Fig. 1d**). We now include these information in the revised main text.

2. In addition to the Western blot quantification of the knockdown efficiency, can the authors also demonstrate the knockdown efficiency using the RNA-Seq data? In addition, is there any evidence of differential isoform expression?

Response: Yes, indeed RNA-seq data also revealed a significant decrease (~ 78%) of STAG2 mRNA upon knockdown (**Supplementary Fig. 7n**), which was similar to what we observed in the RT-PCR and Western blot experiments. As for various STAG2 isoforms, the levels of all major isoforms M14 cells appeared to decrease upon KD (**Rebuttal Table 1**).

Rebuttal table1: Levels of major STAG2 isoforms in M14 WT and KD cells from RNA-seq analysis

Transcript ID	FC	WT_mean	KD_mean
ENST00000218089.9	101.85	196.58	0
ENST00000371160.1	48.1	91.81	0
ENST00000371145.3	40.2	76.41	0
ENST00000371157.3	3.66	748.67	202.66
ENST00000371144.3	2.99	65.03	20.39
ENST00000354548.5	1.62	26.86	15.81
ENST00000455404.1	1.5	29.67	19.03

3. Globally the ChIP-seq replicates appear to be very similar, but what about the differential STAG2 sites? Do they replicate?

Response: As the reviewer suggested, we performed similar analysis to identify loss of STAG2 binding sites for each biological replicate pair (WT and KD) separately. There were 52,139 differential sites found from the sample pair #1, and 36,918 in the sample pair #2, with 34,789 sites common from both pairs (**Supplementary Fig. 1c**), which is close to the number of significant loss of binding sites (32,647) previously identified by merging two biological replicates, suggesting that differential STAG2 sites also replicated between two pairs of samples.

4. Does the change in intensity of the differential STAG2 sites correlate with changes in gene expression using the RNA-Seq data? Also, a volcano plot of the differential gene expression analysis, with the location of IRF9, would benefit the paper.

Response: We thank the reviewer for this important comment. We extracted and analyzed genes with both significant differential expression and significant differential STAG2 peaks at promoters. However, we did not find a clear overall correlation between changes of STAG2 binding at promoters and changes in the corresponding gene expression, as shown in the **Rebuttal Figure 1** below. These results suggest that the change in the STAG2 binding at the promoter alone is not sufficient to account for the change in gene expression. In addition, we have now included a volcano plot of the differential gene expression analysis, as the reviewer suggested, and highlighted IRF9 on the plot (**Supplementary Fig. 7c**)

Rebuttal Figure 1. No significant correlation between STAG2 differential binding site at gene promoter and gene expression. X axis represents fold change of gene expression, and Y axis represents significant gain or loss of STAG2 binding at the corresponding promoter. The lower panel is a zoom-in version of the upper panel.

5. Do the changes observed regarding the TADs in the HiC data correlate with changes in gene expression from the RNA-Seq data?

Response: We thank the reviewer for this important comment. We carried out meta-analysis of gene expression with 10kb sliding window across all variable and stable TADs and their boundaries. Indeed, we observed dramatic changes of gene expression at the boundaries of differential TADs, but not the non-differential TADs (**Supplementary Fig. 4f**). Interestingly, we noted there was an asymmetric pattern for the gene expression change. Gene expression tended to go up at the TAD left (start) sites (in WT, **Fig. 1h**) of variable TADs, but not the TAD right (end) sites (in KD, **Fig. 1h**). The underlying mechanism for this asymmetry pattern remains unknown and will be investigated in the future. We have now included a description of this analysis in the main text.

6. Since the reported findings regarding TADs from the HiC analysis differ somewhat from published findings. The authors should discuss these results. For example, the results of a recent paper by Richart et al. (2021) NAR, which included two shRNAs against STAG2 and HiC replicates, report within TAD changes in chromatin interactions, but little changes affecting compartments and/or boundaries.

Response: Thank you for the suggestion. We have now cited this new NAR paper in the discussion.

7. The TRC clone ID, TRCN0000152538, referenced by the authors (referred to as shSTAG2#38) in the methods does not target STAG2, but DIS3L. Can the authors clarify if this is or is not the correct shRNA?.

Response: We thank the reviewer for pointing out this error. The information on shSTAG2#38 was included by mistake in the original version. In this study, we only used the STAG2 doxycycline-inducible shRNA#60 for STAG2 knockdown, which was previously characterized in our 2016 Nature Medicine paper (Ref #10). We did not use shSTAG2#38 at all in this study. In addition, the TRC clone ID of sh38 should be TRCN0000151938. We have now removed the information on shSTAG2#38 all together from the revised manuscript.

Reviewer #2

Major points:

I find the Hi-C analysis somewhat unconvincing and lacking in important methodological details that would allow readers and reviewers to assess it fully. I realise that resources can limit the number of replicates and sequencing depth, however different parameters (i.e. a bin size appropriate for the sequencing depth) or alternative visualisations could clarify the results.

- The Hi-C resolution used for each analysis or visualization should be specified in the text and/or figure legends, as well as in the methods. The methods currently only mention a resolution of 1Mb for compartment analysis.

Response: We have now included more information on the Hi-C resolution in the figure legends and methods as suggested.

- The insulation score analysis potentially strengthens the findings, however it's unclear how to interpret this as the methods section for this is missing. Typically a more negative insulation score indicates a stronger boundary, however here the scores are positive. If the authors have multiplied the scores by -1 they should say so.

Response: We have now clarified the method for the insulation score analysis and revised the Figure panel (**Supplementary Fig. 5b**). Briefly, we obtained the insulation score by using hicFindTADs at 1 Mb resolution from the HiCExplorer tools.

- I find the examples of TAD boundary loss / gain in Supplementary Fig 5 unclear. It looks like there are changes in interaction strength within TADs in the KD/WT panel, but not any clear examples of boundary loss/gain. Showing the insulation scores or TAD boundary scores alongside the matrices, or highlighting the supposedly changing boundaries with an arrow, might make it clearer. I would also suggest using a larger bin size, as at the current resolution the visualization is quite noisy – a lower resolution might be clearer.

Response: We thank the reviewer for the suggestion. We have tested different resolutions and found better visualization on the change of TADs at 250 kb. We have now included a new figure panel (**Supplementary Fig. 5c**) to show examples of TAD boundary changes, including insulation scores together with the matrices.

- Supplementary Fig 8 focusing on the IRF9 TAD is also unclear, although here at least the difference in TAD structure is visible. It's difficult to understand the positional relationship between the data shown in different panels as some have genomic coordinates and some have gene tracks, but none have both presented legibly.

Response: We thank the reviewer for the suggestion. We have now included a new contact metrics heatmap at 100 Kb bin size, plotted with insulation scores and genomic

coordinates (**Supplementary Fig. 8c**). At this resolution, we can observe a clear expanding of the TAD boundary from the right side of IRF9 after STAG2 KD. The genomic locations of IRF9 and GMPR2, which are the two ends of the newly formed interaction identified by HiChIP, are highlighted.

• I still think it would be helpful to show some visualization, not just a schematic, of the Hi-C data at this locus in a main panel. Perhaps virtual 4C from the IRF9 promoter would make the difference in boundary position visible? Or, again, a lower resolution might actually make the difference in the Hi-C data clearer.

Response: As indicated in the last response, we have now used a 100 Kb bin size for the new Hi-C contact metrics at the IRF9 locus region (**Supplementary Fig. 8c**), which indeed made the difference in the Hi-C data clearer. Due to the space limitation in the main Figure 3, we opt to include this panel in the Supplementary Figure, but we open to further suggestions from the reviewers and the editor.

As noted by another reviewer, the authors should focus on their novel results and tone down the claim (page 9) that the role of STAG2 in 3D genome organization and transcriptional control is novel as this has been previously shown by e.g. Kojic et al 2018, Viny et al. 2019.

Response: We have further toned down the claim in the discussion as the reviewer suggested. Both studies of Kojic et al 2018 and Viny et al. 2019 have been cited.

Minor comments:

The text “difference in TAD boundary’s length” is still in Fig 1f. I think this should be “difference in TAD size”, based on the updated main text.

Response: We have now corrected this error in the Y axis label of **Fig. 1f**.

Fig 1h is now clear to me, but only after seeing the schematic included in the response to reviewers. This should be included in the main figure.

Response: We have now included the schematic diagram in **Fig. 1h**.

At the bottom of page 5, the phrase "the switch/non-switch pattern" is not clear. Is it switch sites or non-switch sites that are significantly associated with TADs, boundaries and enhancers?

Response: We have changed the text to “We found that switch sites were enriched in the boundaries, but depleted in TADs and enhancers, and no significant differences were observed in promoters.”

While the authors note in the response to reviewers that the fact that the majority of altered loops occur within TADs is not statistically significant, this should also be mentioned in the text, otherwise it sounds like an unexpected finding.

Response: We have now including this information in the main text as suggested.

REVIEWERS' COMMENTS

Reviewer #1 (Remarks to the Author):

The authors have sufficiently addressed all of my concerns.

Reviewer #2 (Remarks to the Author):

The updated manuscript is significantly clearer, now that the authors have included the additional details requested by myself and the other reviewer within the main text and figure legends. The resolution of the Hi-C data severely limits the strength of the conclusions that can be drawn from it, but the authors do now include sufficient details on this and other analyses to allow the reader to assess the strength of the evidence for themselves. In addition, the revised discussion better places this work in context with previous work. Therefore, I am satisfied that the manuscript is suitable for publication.